# Demystifying Action Space Design for Robotic Manipulation Policies

**Yuchun Feng** [* 1]  **Jinliang Zheng** [* 1 2]  **Zhihao Wang** [1 3]  **Dongxiu Liu** [1]  **Jianxiong Li** [1]  **Jiangmiao Pang** [2]
**Tai Wang** [2]  **Xianyuan Zhan** [1]

## Abstract

The specification of the action space plays a pivotal role in imitation-based robotic manipulation policy learning, fundamentally shaping the optimization landscape of policy learning. While recent advances have focused heavily on scaling training data and model capacity, the choice of action space remains guided by ad-hoc heuristics or legacy designs, leading to an ambiguous understanding of robotic policy design philosophies. To address this ambiguity, we conducted a large-scale and systematic empirical study, confirming that the action space does have significant and complex impacts on robotic policy learning. We dissect the action design space along temporal and spatial axes, facilitating a structured analysis of how these choices govern both policy learnability and control stability. Based on **13,000+** real-world rollouts on a bimanual robot and evaluation on **500+** trained models over four scenarios, we examine the trade-offs between absolute vs. delta representations, and joint-space vs. task-space parameterizations. Our large-scale results suggest that properly designing the policy to predict delta actions consistently improves performance, while joint-space and task-space representations offer complementary strengths, favoring control stability and generalization, respectively.

## 1. Introduction

Learning-based robotic manipulation policies have achieved remarkable progress in recent years, evolving from simple pick-and-place to dexterous, precision-critical tasks (Brohan et al., 2022; 2023a; Chi et al., 2023; Zhao et al., 2023;

Jang et al., 2022; Zheng et al., 2025b). While recent advances largely focus on scaling training data and model capacity (NVIDIA et al., 2025; Black et al., 2025; Lin et al., 2025a), the specification of *action space*, which is the underlying interface bridging neural predictions and physical hardware, remains an overlooked yet critical determinant of success. As the primary supervision signal, the choice of action representation governs not only the learnability of the policy but also the stability of deployment (Eßer et al., 2024; Zheng et al., 2025a). Subtle changes in this interface can drastically alter the optimization landscape, or even distinguish a robust policy from one that fails to generalize (Chi et al., 2023).

Although important, as illustrated in Figure 1(a), the research community still has no consensus on the best practices of action space design over the past years. Historically, end-effector pose was favored for its semantic simplicity (Liu et al., 2024), yet recent trends have pivoted toward joint-space representations to bypass the numerical instabilities of *inverse kinematics* (Black et al., 2025; Chen et al., 2025). Crucially, the design space extends far beyond simple spatial parameterization. It also involves a combinatorial explosion of choices, including temporal representation (absolute vs. relative) and prediction horizons (e.g., action chunking). Currently, the field lacks a consensus or a unified understanding to navigate through these numerous choices. Researchers often rely on ad-hoc heuristics or legacy configurations inherited from different codebases, leading to a fragmented landscape where "state-of-the-art" results are often conflated with specific, undocumented control choices (Bjorck et al., 2025). Such ambiguity not only impedes reproducibility but also hampers the development of foundation models capable of cross-embodiment transfer.

Addressing this ambiguity is crucial for guiding the design of future robotic manipulation policies. While prior works (Chi et al., 2023; Eßer et al., 2024) have provided some initial insights, deriving comprehensive and reliable guidance for action space selection remains a non-trivial challenge. The difficulty stems from the substantial heterogeneity of robotic learning settings, the limited fidelity of simulation environments, and the high cost of real-world robotic evaluation. Consequently, existing studies are often limited in their empirical scope and lack systematic compar-

---

[*]Equal contribution  [1]Institute for AI Industry Research (AIR), Tsinghua University [2]Shanghai AI Lab [3]Peking University. Correspondence to: Tai Wang <taiwang.me@gmail.com>, Xianyuan Zhan <zhanxianyuan@air.tsinghua.edu.cn>.

*Proceedings of the 43rd International Conference on Machine Learning*, Seoul, South Korea. PMLR 306, 2026. Copyright 2026 by the author(s).

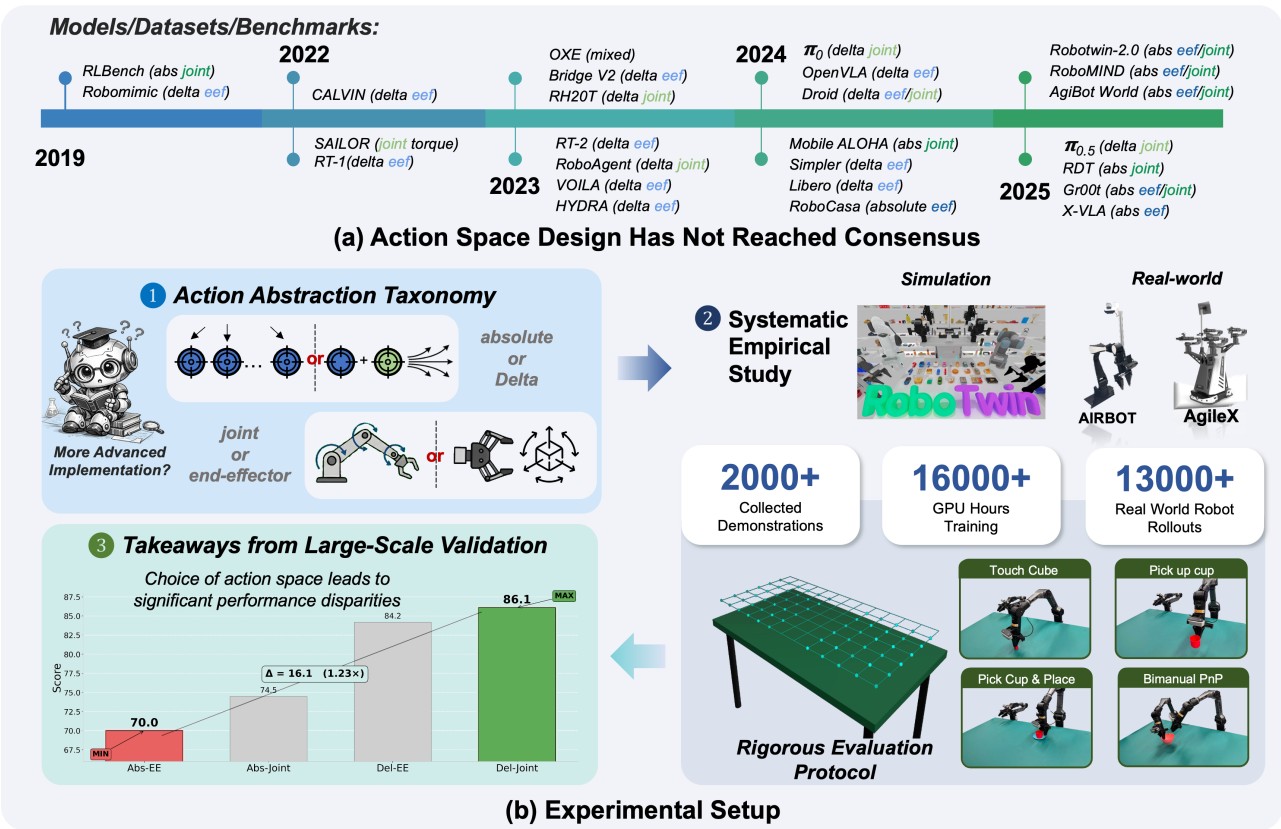

*Figure 1.* Overview of our study on action space design. (a) Historical analysis shows the divergent usage of action spaces (Absolute vs. Delta, Joint vs. EEF) in existing literature. (b) Our experimental setup includes an action abstraction taxonomy and a large-scale benchmark on both simulation and real-world platforms. We invest over 13,000 real-world rollouts to quantify the impact of these design choices, revealing significant performance gaps and identifying best practices for robotic manipulation under various scenarios.

isons. As the field advances toward large-scale generalist robot models (Team et al., 2025), the cost of suboptimal control interface design becomes increasingly consequential, underscoring the need for principled and unified design guidelines. To bridge this gap, we present the first large-scale, systematic empirical study that investigates how action space design impacts robotic policy learning. We begin by formalizing and dissecting action space design along two orthogonal axes: a temporal axis (absolute vs. delta parameterization, action chunking) and a spatial axis (joint-space vs. task-space control). We investigate how these design choices induce fundamental trade-offs between policy learnability, control stability, and deployment performance.

Building on this foundation, we conduct a comprehensive, large-scale experimental study across three platforms: the recently advanced simulation RoboTwin-2.0 (Chen et al., 2025) and two real-world robotics platforms, AgileX PiPER and AIRBOT, all under stringent evaluation protocols. Specifically, the main experiments focus on AgileX, where we design a benchmark suite consisting of four diverse tasks, ranging from precision-critical sanity checks to complex dynamic bimanual coordination. To ensure statistical significance and fair comparison, we introduce a grid-based spatial coverage strategy that standardizes initial conditions across all trials. Through extensive real-world experiments, we first identify the most effective implementation details for each action space design via targeted preliminary studies. We then perform controlled grid searches along several critical axes, including data scale, model expressiveness, and training duration. Overall, this study constitutes a substantial empirical undertaking, comprising over **2,000** collected demonstrations and more than **13,000** real-world rollouts across **500+** trained models. The main results, together with five carefully designed cross-validation experiments, reveal two core insights:

**Summary**

**Temporal abstraction is decisive:** *delta-based* action representations consistently outperform *absolute* actions across all learning paradigms when implemented with appropriate modern practices.

**Spatial abstraction is setting-dependent:** *Joint-space* control excels in standard scenarios with sufficient data, extensive training, and strong model capacity, while *task-space* representations demonstrate superior performance in generalized settings like cross-embodiment and transfer learning.

## 2. Action Abstraction Taxonomy

We consider the problem of imitation learning for robotic manipulation, formulated as learning a policy $\pi_\theta$ from expert trajectories $\mathcal{D} = \{\tau_j\}_{j=1}^M$, $\tau_j = \{(o_t, a_t)\}_{t=1}^{N_j}$ that produces executable actions $a_t$ conditioned on observations $o_t$ at timestep $t$. While this formulation is standard (Li et al., 2025b), the physical realization of an action can vary substantially depending on the control interface exposed to the policy. To reason about these differences in a principled manner, we decompose the action representation space along two orthogonal axes: *spatial abstraction* and *temporal abstraction*, as illustrated in Figure 2. We begin by formalizing these abstractions and their impact on learning. Subsequently, we examine *action chunking*, a pivotal technique in modern architectures, and its interplay with action space design (See Appendix G for detailed formalizations).

### 2.1. Spatial Abstraction

*Spatial Abstraction* defines the abstraction boundary between the learned policy and the hardware controller. At the lowest level of spatial abstraction lies the actuator space (e.g., motor torques or currents), which directly governs the robot's physical dynamics. However, direct torque-level supervision is rarely adopted for high-level manipulation policies, as it suffers from high dimensionality and poor sample efficiency (Yu et al., 2025). Accordingly, we restrict our scope to the two dominant kinematic abstractions used in practice: *configuration space* (joint positions) and *task space* (robot-based end-effector pose). While joint space and task space are kinematically equivalent via forward and inverse kinematics solvers, the choice of supervision domain induces distinct optimization landscapes. Task-space control provides a geometrically meaningful abstraction that aligns naturally with object-centric visual observations. However, it relies on inverse kinematics solvers during deployment, which introduces numerical singularities and error accumulation that can degrade execution robustness (Lee, 1982). In contrast, joint-space control avoids solving inverse kinematics, but this robustness comes at the cost of increased learning complexity: the policy must implicitly learn the robot's kinematic structure, mapping visual inputs onto a highly non-linear configuration manifold. Consequently, spatial abstraction presents a fundamental trade-off between learning alignment and execution robustness.

### 2.2. Temporal Abstraction

The second axis of analysis, orthogonal to the spatial axis, is *temporal abstraction*, which specifies the order of temporal derivatives represented by the predicted action sequence. At one end of the spectrum is *absolute* representation ($0^{\text{th}}$-order), which specifies target states directly. At the other end are *relative* or *delta* representations ($1^{\text{st}}$-order), which specify state increments. It is worth noting that we adopt a position-based low-level controller as the interface, aligning with standard practices in the community (Black et al., 2025). Consequently, our "$1^{\text{st}}$-order" formulation refers to the *semantic meaning* of the policy output rather than the physical control mode, decoupling high-level motion planning from low-level dynamic regulation. Although higher-order formulations like force control are possible, they generally rely on accurate inertial modeling and significantly increase system complexity (Yu et al., 2025). Consequently, we focus our analysis on absolute ($0^{\text{th}}$-order) and delta ($1^{\text{st}}$-order) representations, which governs a fundamental trade-off between learning stability and control accuracy. Under the absolute parameterization, the policy must map observations to global target states. This interface can encourage intuitive and precise grounding; however, it also requires the model to internalize complex real-world geometry and to cope with highly variable target distributions (Chi et al., 2023; Liu et al., 2025), thereby inducing substantial learning difficulty. In contrast, a delta parameterization predicts relative increments, yielding a better-conditioned and closed-loop learning target. However, deploying delta actions makes the system more sensitive to feedback imperfections: noise, latency, and tracking errors can accumulate over time and lead to drift (Zhang et al., 2025a).

### 2.3. Action Chunking

Throughout this work, we adopt *Action Chunking* (Zhao et al., 2023) as a default component. This technique has emerged as a cornerstone for action space shaping. By predicting a sequence of future actions, policies can better capture temporal dependencies, leading to substantial performance gains (Zhang et al., 2025a). However, we identify that the introduction of chunking creates a non-trivial structural ambiguity, resulting in two critical design challenges that remain under-explored in the robotics literature:

**1. Ambiguity in Delta Alignment.** Integrating chunking with delta actions necessitates a choice of reference frame: *step-wise delta* (relative to the immediately preceding predicted state within the sequence) (Liu et al., 2024; Mees

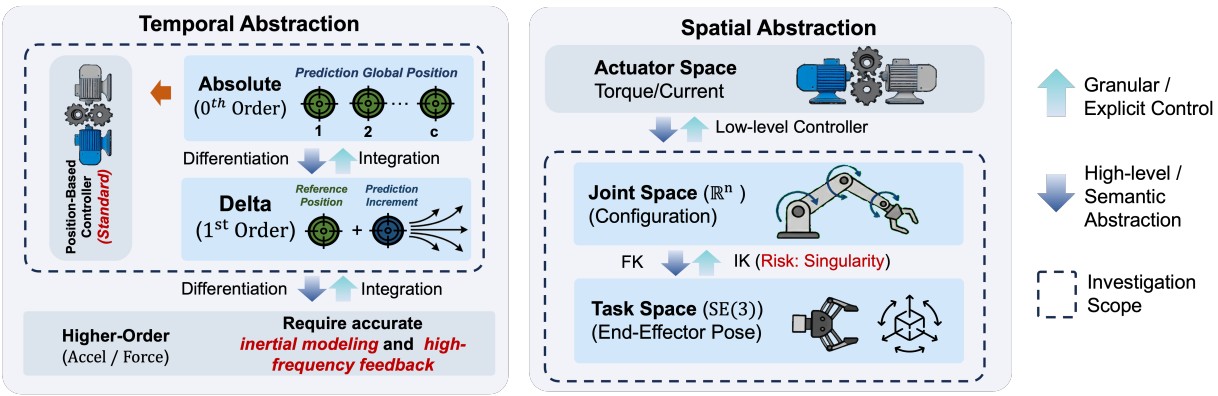

*Figure 2.* Hierarchy of the action space for robotic manipulation policies and its abstraction taxonomy

et al., 2022) versus *chunk-wise delta* (relative to the robot's state at the start of the chunk) (Black et al., 2025). This choice fundamentally reshapes the action distribution.

**2. Horizon-Abstraction Coupling.** While the chunking horizon $k$ is typically optimized as an isolated hyperparameter, we hypothesize a fundamental coupling between $k$ and the choice of action abstraction. Specifically, delta-based control may necessitate shorter horizons to facilitate rapid correction, whereas absolute position control might benefit from longer horizons to maintain global spatial grounding. Elucidating the interplay between these factors is a prerequisite for designing a proper action space for policy learning.

## 3. Experimental Setup

The goal of this study is to derive **practical and generalizable guidelines** for action space design. To ensure that these guidelines are robust across different scenarios, we conduct a large-scale empirical investigation spanning multiple hardware platforms, task configurations, and learning regimes. In this section, we summarize the common experimental components used throughout the paper, including model variations, hardware setups and evaluation protocol.

### 3.1. Model Architectures for Policy Learning

We aim to establish guidelines across a spectrum of paradigms, from specialized architectures like ACT (Zhao et al., 2023) and Diffusion Policy (Chi et al., 2023), to foundation models like $\pi_0$ (Physical Intelligence et al., 2025).

Following common practice (Brohan et al., 2022), we design a base architecture for policy learning. It uses a FiLM-conditioned ResNet-18 vision encoder (Perez et al., 2018) paired with a 6-layer Transformer decoder. Further details of the implementation are provided in Appendix D. To investigate the interplay between action space design and model expressiveness, we adopt two prominent generative modeling paradigms, resulting in the following two model variants, which correspond to implementation in ACT (Zhao et al., 2023) and DP (Chi et al., 2023), respectively: **1. Regression-**

based Policy is optimized using a standard Mean Squared Error (MSE) loss:

$$\mathcal{L}_{\mathrm{R}} = \mathbb{E}_{(\mathbf{o},\mathbf{a})\sim\mathcal{D}} \left[ \left| \pi_\theta(\mathbf{o}) - \mathbf{a} \right|^2 \right],$$

**2. Flow Matching-based Policy** provides more powerful modeling for complex distributions by learning a velocity field $v_\theta$ that transforms noise $\epsilon$ into the expert action $\mathbf{a}$:

$$\mathcal{L}_{\mathrm{F}} = \mathbb{E}_{\tau\sim\mathcal{U}(0,1),\,(o,a)\sim\mathcal{D}} \left[ \left\| v_\theta(a^\tau, o, t) - (a - \epsilon) \right\|^2 \right],$$

where $\mathbf{x}_\tau = (1 - \tau)\epsilon + \tau\mathbf{a}$, and $\tau \sim \mathcal{U}(0,1)$. Common detailed training setups are provided in Appendix D, while specific learning regimes are discussed independently in the subsequent analysis sections. We further introduce a **Foundation Policy** ($\pi_0$), designed specifically to investigate the transfer learning properties across different action spaces. Further discussion for action space design with pretraining-finetuning paradigm is provided in Appendix B.

### 3.2. Robotic Platforms and Evaluation Protocol

Our experiments are conducted across four distinct robotic hardware configurations: (1) a single-arm AgileX platform serving as the primary setup for large-scale real-world experiments; (2) a dual-arm AgileX platform and (3) a single-arm AIRBOT platform, both of which are utilized to evaluate cross-platform generalizability; and (4) RoboTwin 2.0, a simulation benchmark designed for large-scale, reproducible experiments under controlled environments. The overview and detailed description of our experimental setup can be found in Figure 1(b) and Appendix E, respectively.

**Evaluation Protocol for Real-World Experiments.** For the real-world experiments, we designed a curriculum consisting of four manipulation tasks: `Touch Cube`, `Pick Up Cup`, `Pick and Place Cup`, and `Bimanual Cube Transfer`. These tasks are characterized by increasing contact richness, temporal horizons, and coordination requirements. Detailed task descriptions are provided

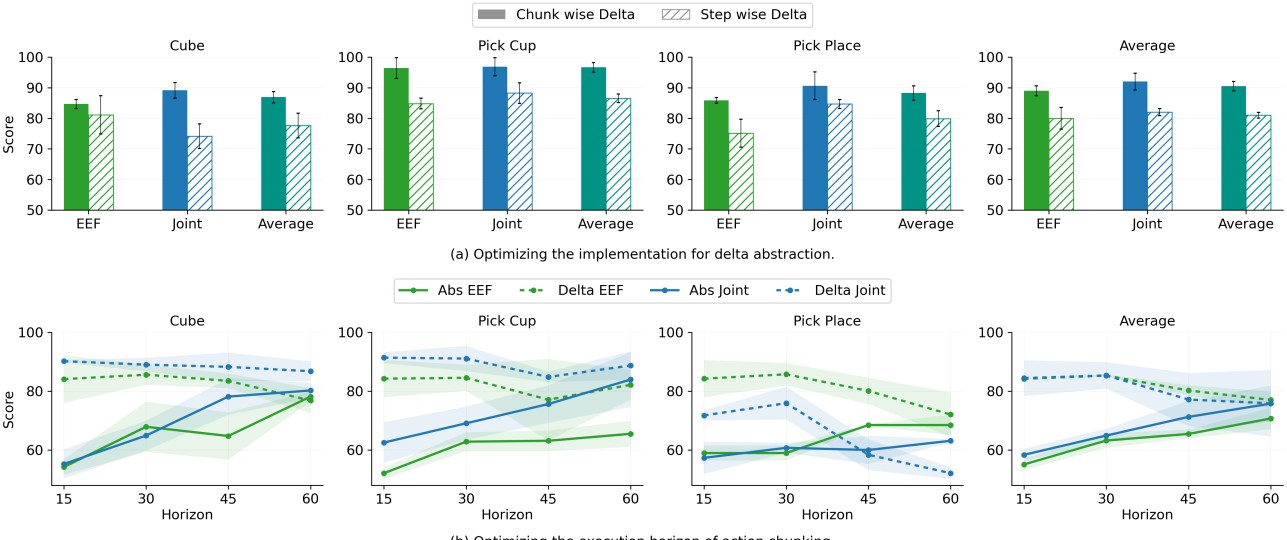

(a) Optimizing the implementation for delta abstraction.

(b) Optimizing the execution horizon of action chunking.

*Figure 3.* (a) We verified that chunk-wise delta for both EEF and Joint perform better than step-wise delta representations. (b) Grid search over execution horizons across four different action space.

in Appendix D. To ensure reproducibility and statistical significance, we implement a rigorous evaluation protocol to ensure spatial coverage. Specifically, the robot's workspace is uniformly partitioned into a $6 \times 6$ grid. Both data collection and testing procedures strictly adhere to this protocol by initializing object positions uniformly across these grids, thereby mitigating potential distribution shifts between training and evaluation. For each real-world experiment, we report progress score based on three independent trials, where each trial comprises 10 individual rollouts.

**Evaluation Protocol for Simulation.** We adopt the AgileX embodiment within RoboTwin 2.0 simulation (hard mode) and select a subset of 10 tasks from the 50 officially provided tasks. Aligned with our real-world evaluation protocol, we report the average success rate across three independent trials, with each trial consisting of 10 rollouts per task.

## 4. Results and Analyses

In this section, we systematically evaluate different action space by addressing three progressive research questions (RQs), moving from foundational implementation nuances to generalizable trends and large-scale robustness:

**RQ1 (Foundational Impact):** *At the implementation level, how do specific choices in action space realization influence policy performance, and what constitutes the optimal configuration?*

**RQ2 (Generalizable Trends):** *Building upon these optimized implementations, can we identify consistent trends across diverse tasks that dictate the selection of superior action abstractions?*

**RQ3 (Systemic Robustness):** *Finally, do the identified trends remain robust when subjected to more advanced settings, such as scaled data regimes, foundation model transfer, and cross-embodiment learning?*

### 4.1. RQ1: Implementation Nuances are Decisive

We begin by addressing the implementation ambiguities introduced by *action chunking* as identified in Sec. 2.3. These implementation nuances are often overlooked but, as we demonstrate, are decisive for policy stability. We provide more implementation nuances in Appendix F.

#### 4.1.1. SUPERIORITY OF CHUNK- VS. STEP-WISE DELTA

We first conduct real-world experiments to evaluate the performance impact of chunk-wise and step-wise delta actions. Fig. 3(a) reports the performance of a standard regression-based policy on a single-arm AgileX platform across three foundational tasks: `Touch Cube`, `Pick Up Cup`, and `Pick and Place Cup`. Extended cross-validation results and training details are available in Appendix F. Our results demonstrate that ***chunk-wise delta* consistently and significantly outperforms step-wise delta across all tasks**. Notably, the performance gap reaches upwards of **10%** on average, a substantial margin that underscores the importance of alignment frame selection.

To explain the underlying mechanism, we analyze the action decoding process under a fundamental stability criterion: ***a robust action representation should not amplify prediction errors during deployment.***

**Proposition 4.1** (Noise Amplification in Step-wise Integration). *Let $\epsilon \in \mathbb{R}^k$ be the prediction noise for a chunk of length $k$, with bounded norm $\|\epsilon\|_2 \leq \delta$. The cumulative*

*Table 1.* Quantitative comparison of progress scores and standard errors across embodiments and tasks. The results contrast Regression (ACT) and Flow Matching (DP) under four distinct control interface configurations. **Bold** and underlined values denote the best and second-best performance for ACT and DP separately.

| Task | EE (ACT) | | Joint (ACT) | | EE (DP) | | Joint (DP) | |
|---|---|---|---|---|---|---|---|---|
| | abs | delta | abs | delta | abs | delta | abs | delta |
| Single Arm AgileX | | | | | | | | |
| Cube | $77.1 \pm 3.8$ | $\mathbf{86.2} \pm 3.5$ | $77.2 \pm 2.8$ | $\underline{84.8} \pm 3.6$ | $83.6 \pm 2.0$ | $91.5 \pm 3.7$ | $\underline{95.5} \pm 2.3$ | $\mathbf{96.7} \pm 1.8$ |
| Pick | $63.1 \pm 3.7$ | $\mathbf{97.9} \pm 2.1$ | $83.9 \pm 9.8$ | $\underline{95.2} \pm 4.8$ | $64.6 \pm 7.5$ | $\mathbf{97.9} \pm 2.1$ | $77.7 \pm 8.6$ | $\underline{97.6} \pm 2.4$ |
| Pick Place | $66.8 \pm 6.5$ | $\mathbf{84.7} \pm 6.4$ | $70.8 \pm 2.8$ | $\underline{83.8} \pm 4.6$ | $73.8 \pm 1.2$ | $\underline{84.8} \pm 1.9$ | $81.9 \pm 4.1$ | $\mathbf{93.5} \pm 0.3$ |
| Average | $69.0 \pm 2.0$ | $\mathbf{89.6} \pm 2.1$ | $77.3 \pm 2.8$ | $\underline{88.0} \pm 2.9$ | $74.0 \pm 3.1$ | $\underline{91.4} \pm 1.6$ | $85.0 \pm 2.3$ | $\mathbf{95.9} \pm 1.1$ |
| Single Arm AgileX (Multi) | | | | | | | | |
| Cube | $\underline{89.7} \pm 5.7$ | $\mathbf{96.4} \pm 3.6$ | $94.4 \pm 3.1$ | $88.8 \pm 3.5$ | $96.4 \pm 3.6$ | $93.2 \pm 4.1$ | $\underline{97.9} \pm 2.1$ | $\mathbf{100.0} \pm 0.0$ |
| Pick | $67.6 \pm 9.2$ | $\mathbf{89.3} \pm 4.0$ | $77.7 \pm 6.9$ | $\underline{95.8} \pm 4.2$ | $\underline{91.1} \pm 2.7$ | $\mathbf{100.0} \pm 0.0$ | $90.8 \pm 6.4$ | $\mathbf{100.0} \pm 0.0$ |
| Pick Place | $52.2 \pm 10.0$ | $72.3 \pm 6.1$ | $\mathbf{73.8} \pm 7.3$ | $\underline{72.8} \pm 2.8$ | $75.0 \pm 5.4$ | $83.5 \pm 2.6$ | $\underline{86.2} \pm 4.3$ | $\mathbf{93.3} \pm 3.4$ |
| Average | $69.8 \pm 1.0$ | $\mathbf{86.0} \pm 3.1$ | $81.9 \pm 4.2$ | $\underline{85.8} \pm 0.9$ | $87.5 \pm 2.8$ | $\underline{92.2} \pm 2.3$ | $91.6 \pm 1.1$ | $\mathbf{97.8} \pm 1.1$ |
| Bimanual AgileX | | | | | | | | |
| Bowl | $63.7 \pm 8.3$ | $\underline{67.0} \pm 5.2$ | $51.6 \pm 7.7$ | $\mathbf{69.6} \pm 5.5$ | $64.6 \pm 7.5$ | $\mathbf{75.3} \pm 9.3$ | $\underline{74.7} \pm 3.9$ | $74.6 \pm 5.2$ |
| RoboTwin 2.0 | | | | | | | | |
| Average | $26.7 \pm 3.3$ | $33.3 \pm 6.1$ | $\underline{40.0} \pm 0.6$ | $\mathbf{46.3} \pm 1.9$ | $26.0 \pm 1.2$ | $\underline{37.0} \pm 6.2$ | $32.3 \pm 2.6$ | $\mathbf{48.0} \pm 4.4$ |
| Overall Avg | $63.4 \pm 2.7$ | $\underline{78.4} \pm 1.4$ | $71.2 \pm 2.9$ | $\mathbf{79.7} \pm 2.5$ | $71.9 \pm 4.8$ | $\underline{82.9} \pm 1.6$ | $79.6 \pm 2.2$ | $\mathbf{88.0} \pm 2.3$ |

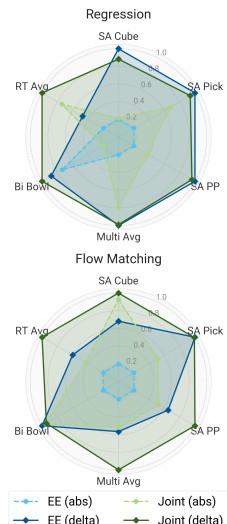

*Figure 4.* Normalized Score Comparison.

*error in the decoded executable actions, denoted as $\mathbf{e}_a$, relates to $\epsilon$ via a linear transformation matrix $\mathbf{M}$:*
*(1) For **step-wise delta**, $\mathbf{M}_{\text{step}} = \mathbf{L}_k$, where $\mathbf{L}_k$ is the $k \times k$ lower-triangular matrix of ones. The worst-case error bound scales linearly with the horizon: $\|\mathbf{e}_a\|_2 \leq \|\mathbf{L}_k\|_2\|\epsilon\|_2 \approx \frac{2k+1}{\pi}\delta \sim \mathcal{O}(k)$.*
*(2) For **chunk-wise delta** and **absolute** action, $\mathbf{M} = \mathbf{I}_k$, implying independent error propagation: $\|\mathbf{e}_a\|_2 \leq \|\mathbf{I}_k\|_2\|\epsilon\|_2 = \delta \sim \mathcal{O}(1)$.*

Proof and detailed analysis see Appendix G.2. As demonstrated in Proposition 4.1, *step-wise* integration inherently amplifies prediction noise as the horizon $k$ increases, whereas *chunk-wise* and *absolute* action **maintain a constant error bound**. This theoretical result corroborates our empirical findings, confirming that chunk-wise delta yields a structurally more reliable representation.

### 4.1.2. INTERPLAY WITH HORIZON $k$

Building upon the optimized chunk-wise implementation, we investigate the chunking horizon $k$. Following the experimental setup described in Sec. 4.1, we conducted a grid search across horizons. Crucially, following the practice in $\pi$ (Black et al., 2025), all policies were trained using a consistent, longer horizon of $k = 60$ (2 seconds at 30 Hz) to ensure maximum supervision efficiency and temporal coherence. During inference, we then grid-searched the execution horizon from 15 to 60 to identify the optimal deployment window for each representation.

The results in Fig. 3(b) reveal an insightful phenomenon: **absolute control benefits from a significantly longer horizon, whereas delta control peaks at a shorter horizon**. This observation aligns with our hypothesis in Sec. 2.3 regarding the sensitivity of relative representations to execu-

tion drift. Notably, in several tasks, we observe a saturation point even for absolute actions, where increasing the execution horizon no longer yields significant gains. This phenomenon suggests a potential **information decorrelation** effect (See Appendix G). Managing the trade-off between the stability provided by long-horizon absolute grounding and the inherent information decay of distant predictions remains a compelling avenue for future research. We provide further discussion in Appendix B.

> **Takeaway**
>
> **Implementation Nuances are Decisive**:
> (1) *Chunk-wise delta* is fundamentally superior to *step-wise delta*. (2) Optimal horizons are critical and abstraction-dependent: *delta* requires shorter windows while *absolute* thrives with longer horizons.

### 4.2. RQ2: Systematic Trends in Action Abstraction

With the implementation strategies optimized in RQ1, we standardize all delta-based actions to the *chunk-wise* alignment frame. In addition, to accommodate the horizon-coupling effect identified in earlier analysis, we employ shorter execution horizons ($k = 30$) for *delta* actions and longer horizons ($k = 60$) for *absolute* actions for optimal performance. With this foundation, we then pivot to the central inquiry: *how do different action space influence performance across diverse embodiments, model variations, and learning regimes?*

To investigate these dimensions systematically, we conducted extensive experiments across three platforms: a single-arm AgileX robot, a bimanual AgileX robot, and the RoboTwin-2.0 simulation environment. Our evaluation

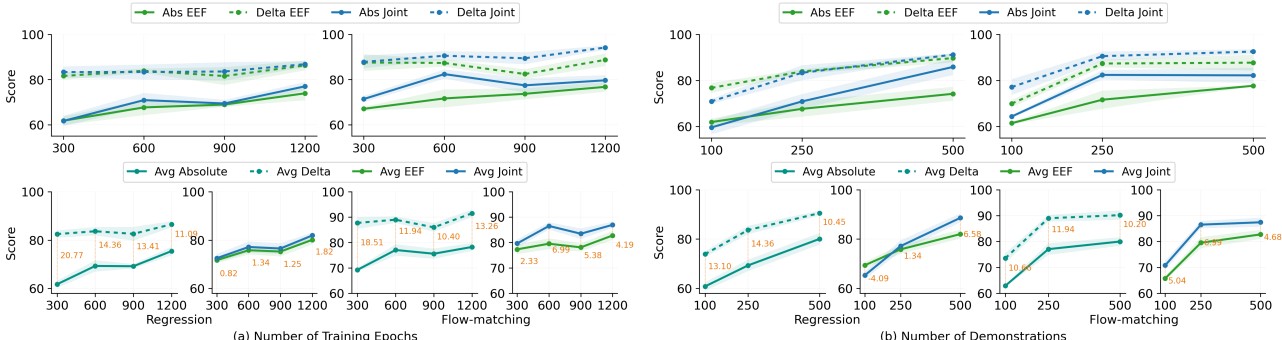

*Figure 5.* **Consistency of Action Space Superiority under Scaling.** We evaluate policy performance across varying (a) training epochs and (b) number of demonstrations. The top row illustrates the individual performance of the four action space, while the bottom row aggregates the performance across temporal (*abs* vs. *delta*) and spatial (*task* vs. *joint*) dimensions, providing an intuitive comparison.

spans 14 distinct tasks in total: 4 real-world tasks and 10 simulation tasks, as described in Sec 3. For data collection, we utilized 250 expert demonstrations per task for real-world experiments and 50 per task for the RoboTwin-2.0. To ensure the generality of our findings, we evaluated both standard regression-based and flow-matching-based policy networks, each trained for 600 epochs. Furthermore, we introduced both single- and multi-task learning settings on the single-arm platform to examine how different abstractions withstand task interference and distribution shifts. Detailed experimental specifications are provided in Appendix E.1.

The comprehensive experimental results are reported in Table 1. Additionally, we provide a radar plot (Fig. 4) with normalized scores to highlight the difference and facilitate an intuitive comparison across different action spaces. In the subsequent sections, we conduct a deep dive into the spatial and temporal abstraction axes, independently analyzing the superiority of various action abstractions.

#### 4.2.1. TEMPORAL ABSTRACTION

We first conduct an in-depth analysis along the **temporal axis**. Even when utilizing the optimal implementation identified for both *absolute* and *delta* abstractions in RQ1, a substantial performance gap remains evident. Specifically, we observe that **with standard modern practice, *delta* abstraction consistently and significantly outperforms *absolute* abstraction across all platforms, task configurations, and model variations.**

We attribute this superiority to two primary factors: (1) As discussed in Sec 2, learning a direct mapping from high-dimensional visual observations to global coordinates presents a significant challenge. Even for policies with strong expressiveness and proper normalization, global coordinates in both *Task* and *Joint* spaces exhibit lower local coherence. In contrast, the properly implemented *chunk-wise delta* actions allow the network to focus on the immediate displacement, which is a more tractable inductive bias. (2) There has been insufficient exploration of the horizon

trade-off, as noted in Sec 2.3. While *absolute* representations prefer a longer execution horizon to maintain global consistency, training policies with long horizons remains a complex challenge.

While we hope these findings encourage further exploration into *absolute* representations to leverage their potential for precise global grounding, the deterministic conclusion from our current empirical study is that ***delta* abstraction provides a more robust and sample-efficient foundation for modern imitation learning backbones.**

#### 4.2.2. SPATIAL ABSTRACTION

Along the axis of spatial abstraction, we observe **an overall performance superiority for actions in *Joint space* compared to *Task space***. However, unlike the deterministic conclusion reached in the temporal analysis, this spatial superiority contains some inconsistency across different platforms, tasks, and learning regimes.

Amidst this complex and entangled data, we identify a particularly insightful phenomenon: **policies trained under the flow-matching generative paradigm exhibit a distinct excellence in *Joint space* learning.** Specifically, as shown in Figure 4, the performance envelope for flow-matching-based models significantly expands when transitioned from *task* to *joint space*. These results are closely aligned with our discussion in Sec 2: as the action distribution in *joint space* often resides on a complex, non-linear, and multi-modal hardware configuration manifold (Bensadoun et al., 2022), powerful generative modeling is required to effectively capture the underlying structure. While standard regression backbones often struggle with such complexity and multi-modality, the flow-matching paradigm excels at modeling these intricate joint-space distributions, thereby unlocking the full potential of intrinsic control stability and preserving the essential kinematic meaning of *Joint space*.

Takeaway

(1) ***Delta actions*** serve as a superior temporal abstraction for modern policy backbones. (2) ***Joint-space control*** generally provides a more robust spatial representation, particularly when paired with **strong generative modeling** (e.g., diffusion).

## 4.3. RQ3: Consistency and Scaling Analysis

In this section, we extend our experiments to broader settings and larger-scale validations under varied control conditions. We investigate whether the conclusions derived in RQ2 hold firm when the learning regime is scaled in terms of data volume and computation budgets. Furthermore, we evaluate these abstractions within advanced learning setups, including transfer learning from pretrained robotics foundation models and cross-embodiment learning scenarios.

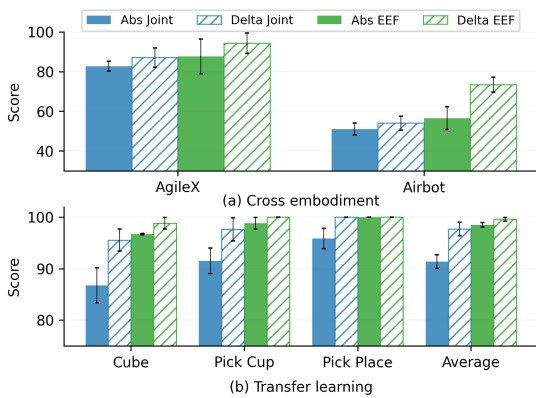

*Figure 6.* Comparison across different action space on advanced learning settings

### 4.3.1. SCALING WITH DATA AND COMPUTE

We extended the RQ2 experiments to a much larger scale, covering all three hardware platforms, learning regimes, and model architectures. To evaluate *Compute Scaling*, we introduced three additional training milestones at 600, 900, and 1200 epochs. For *Data Scaling*, we varied the demonstration volume (100, 250, and 500 trajectories) for real-world experiments. The results of single-task learning on the AgileX platform across the four designed real-world tasks, aggregated in Figure 5, provide a high-fidelity map of action abstraction performance across varying resource budgets. Each data point in the figure represents the averaged outcome of 12 independent validation trials across different setups (totaling 120 real-world rollouts per point). Further exhaustive cross-validation is presented in Appendix F.

Crucially, these results substantiate a definitive trend: *delta actions* consistently serve as the superior temporal abstraction, albeit with marginal gains in certain settings, while *joint-space actions* remain the most robust spatial representation in most cases. Notably, as training epochs and

data volume increase, the superiority of *joint-space* actions becomes increasingly pronounced, particularly for regression-based policies. This mirrors our observations in RQ2: while *task-space* can be competitive in low-data or limited-compute regimes, *joint-space* benefits disproportionately from stronger modeling capabilities and extensive training. This suggests that *joint-space* representations are **fundamentally better suited to capturing the underlying kinematic manifold as the learning regime scales**.

### 4.3.2. ADVANCED LEARNING REGIMES

To align with increasing learning demands and the objective of pursuing a more generalizable embodied agent, this section examines two prominent learning regimes: cross-embodiment learning and transfer learning from foundation models. Specifically, we introduce a new embodiment, AIRBOT, for cross-embodiment learning and employ $\pi_0$ as the foundation model to perform transfer learning. We report the average cross-embodiment performance across the AgileX and AIRBOT platforms in Figure 6(a), and the transfer learning results across three single-arm tasks on AgileX in Figure 6(b). See Appendix E.3 for experimental details.

The results again confirm the superiority of *delta control* compared to *absolute actions*. However, regarding spatial representations, we observe an interesting shift: under cross-embodiment and transfer learning settings, *task-space* representations exhibit a more pronounced advantage and, in some cases, surpass *joint-space* control. We attribute this behavior to the relatively embodiment-invariant nature of *task-space* representations, which abstract away robot-specific kinematics and thereby facilitate knowledge transfer across different embodiments. These results highlight a complementary strength of *task-space* control in scenarios where generalization across robots or tasks is prioritized over execution robustness within a fixed embodiment.

Takeaway

(1) The superiority of ***delta actions*** remains consistent across diverse learning regimes. (2) ***joint-space actions*** benefit exceptionally from stronger modeling capacity and extensive training. (3) ***task-space representations*** demonstrate a pronounced advantage in generalized settings, specifically within cross-embodiment and transfer learning regimes.

## 5. Conclusion and Practical Implications

In this work, we presented a systematic evaluation of action space design for IL-based robotic manipulation, decomposing the problem into orthogonal spatial and temporal axes. By conducting large-scale experiments across 13,000+ real-world rollouts and extensive testing in simulation environment, we establish that action space is far from a trivial

implementation detail. Instead, our results reveal that action representation is a decisive configuration that interacts non-trivially with diverse learning regimes. We summarize our findings into the following actionable guidelines to standardize future research and deployment:

1. The execution horizon $k$ of action chunking should not be treated as an isolated constant but must be adapted to the temporal abstraction.

2. For standard imitation learning settings with sufficient resources where the primary objective is to maximize performance on a specific hardware platform (e.g., single-arm manipulation), the combination of ***joint space*** and ***chunk-wise delta*** yields the most robust results.

3. When the objective shifts towards generalized setting like cross-embodiment or transfer learning, ***task space*** (**EE**) becomes the superior spatial abstraction.

Despite the unprecedented scale of this study, our derived insights represent a first step towards a unified understanding of action spaces. We highlight several exciting directions that remain to be explored in Appendix B, paving the way for future research.

## Impact Statement

This paper presents work whose goal is to advance the understanding of robotic manipulation policy learning through large scale empirical analysis of action space design. We believe this work may help improve the stability, reproducibility, and effectiveness of future robot learning systems. Since this work focuses on foundational empirical evaluation rather than deployment of autonomous robotic agents, we do not identify immediate harmful societal consequences directly arising from this research.

## Acknowledgement

This work is funded in part by the National Key R&D Program of China (2022ZD0160201). This work is also supported by Wuxi Research Institute of Applied Technologies, Tsinghua University under Grant 20242001120, Shanghai Artificial Intelligence Laboratory, Xiongan AI Institute, and SunRisingAI Lab.

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

## A. Ethics and Reproducibility Statement

In this paper, we employed Large Language Models (LLMs) **solely** for polishing the writing. No parts of the technical content, experimental results, or conclusions were generated by LLMs. All real-world data utilized in this study were collected using our custom hardware platform following the protocols detailed in the paper. Simulation data were generated using open-source environments. To support reproducibility, all code and datasets will be **open-sourced** upon publication. We confirm that our datasets contain no personally identifiable information and pose no risks regarding privacy violations or social bias.

## B. Limitations and Future Work

In this section, we look beyond the scope of our current benchmarking to identify several prospective directions for action space design. While our study establishes a strong baseline, we believe the following areas hold significant promise for unlocking the next generation of robotic control.

**1. Beyond Rigid Taxonomies: Hybrid and Adaptive Representations.** Our current analysis operates within a fixed taxonomy (e.g., strictly *Absolute* vs. *Delta*, or *Joint* vs. *Task*). However, the optimal action space may not be static. A compelling avenue for future work is to explore **hybrid or adaptive action spaces** that dynamically switch representations based on the task phase. For instance, a policy might utilize *task-space delta* actions for the reaching phase to maximize generalization, while seamlessly transitioning to *joint-space absolute* actions for fine-grained manipulation or contact-rich stages. Learning to modulate these representations end-to-end could theoretically combine the stability of global grounding with the local precision of relative control. Another significant but under-explored dimension is the theoretical underpinning of *Action Chunking*. While Sec. 2.3 addresses the practical implementation nuances, our understanding of this mechanism remains empirical. Current strategies for selecting chunking horizons are heuristic and likely suboptimal. For instance, while our results demonstrate that extending the training horizon benefits *absolute actions*, training with longer horizons introduces distinct, under-explored challenges regarding convergence stability and information decorrelation. A critical gap remains in rigorously formalizing **how action chunking reshapes the optimization landscape** to derive principled, rather than heuristic, horizon selection strategies.

**2. Scaling to High-DoF Morphologies, Dynamic and Dexterous Tasks.** While our experiments cover several standard platforms for robotic manipulation, the intrinsic connection between action space and **morphological complexity** warrants deeper investigation. It remains an open question whether our findings—specifically the superiority of delta joint actions—generalize to high-degree-of-freedom systems, such as humanoids or multi-fingered hands, where the kinematic manifold exhibits significantly higher dimensionality and non-linearity. Furthermore, extending this evaluation to highly dynamic or dexterous domains, such as table tennis or deformable object manipulation (e.g., cloth folding), could uncover critical constraints on action latency and horizon coupling that are less visible in pick-and-place tasks.

**3. Unifying Action Spaces for Generalization and Transfer.** In Sec. 4.3, we conduct a preliminary investigation into the interplay between action space design and cross-embodiment generalization. Our results, spanning cross-embodiment learning across distinct morphologies and transfer from the $\pi_0$ foundation model, demonstrate a **pronounced superiority** of *task space (EE)* control. This advantage is primarily attributable to its inherent embodiment invariance. Nevertheless, this observation requires broader validation. A critical next step is to extend the analysis to a wider range of embodiment pairs and foundation models, with particular attention to how different pretraining paradigms, such as vision language alignment and pure behavioral cloning, interact with action space selection. More specifically, an important open question is whether aligning pretraining supervision directly in *joint space* can reduce the transfer gap and potentially challenge the current dominance of task-space representations in foundation models.

## C. Related Work

**Learning-Based Robotic Manipulation Policies.** Robotic manipulation policies have made remarkable progress in recent years, advancing from simple atomic pick-and-place tasks (Kim et al., 2024; Brohan et al., 2022; 2023a; O'Neill et al., 2023), to long-horizon sequential tasks (Du et al.; Brohan et al., 2023b; Fang et al., 2025; Shi et al.), fine-grained contact-rich manipulations (Chi et al., 2023; Yu et al., 2025; Zhang et al., 2025b), and even complex dexterous skills (Zhao et al.; Physical Intelligence et al., 2025; Bjorck et al., 2025; Team et al., 2025; Zhao et al., 2023). These policies are typically formulated as end-to-end models: given images, robot proprioceptive states, and optionally language prompts or other modalities such as point clouds (Li et al., 2025a; Xue et al.; Ze et al., 2024) or force feedback (Zhang et al., 2025b; Yu et al.,

2025), the model must generate *actions* that directly control the robot. Auxiliary supervision signals, such as future image prediction (Cheang et al., 2024; Bjorck et al., 2025; Cheang et al., 2025; Ye et al., 2024; Li et al.; Wen et al., 2023; Liu et al.), object detection (Zheng et al., 2024; Team et al., 2025), or sub-language planning (Black et al., 2025; Shi et al.; Lin et al., 2025b), are introduced to leverage web knowledge and improve generalization. However, *ACTION* is the only modality that can be the interface for robotics models to interact with the 3D world, and thus is the indispensable modality for robotic learning that ultimately governs execution performance, making it one of the most critical design choices in policy learning (Lee et al., 2024; Chi et al., 2023).

**Chaotic Control Interfaces for Robotic Policies.** The physical representations of actions vary widely, providing numerous options for supervision. A fundamental choice is whether to represent actions in joint space that directly control motors, or in End-Effector (EEF) space, which specifies the gripper's position and orientation in 3D space (Doshi et al., 2024). At first glance, these two representations may seem interchangeable, since forward kinematics (FK) maps joints to EEF poses and inverse kinematics (IK) provides the reverse mapping. However, as we show in this paper, joint-based and EEF-based actions exhibit markedly different training behaviors and preferences. Furthermore, actions can be parameterized in different orders, such as 0th-order positions (Liu et al., 2025; Chi et al., 2023; Zhao et al., 2023), 1st-order velocities (Team et al., 2024; Doshi et al., 2024), or even higher-order derivatives, adding further variability. To date, no clear consensus has emerged on which representation is most effective, where most different works adopt different interfaces without concrete reasons (Chi et al., 2023; Zhao et al., 2023; Physical Intelligence et al., 2025; Doshi et al., 2024; Chi et al., 2024).

## D. Model Implementation and Training Details

In this section, we provide more details about the model implementation and training procedure. The overview of our implemented model architecture is illustrated in Fig 7. Following prior works (Liu et al.; Zhao et al., 2023; Chi et al., 2023), our model comprises a **FiLM-conditioned ResNet-18** backbone that injects language features into visual representations, and a **Transformer-based encoder-decoder** for action generation. Our architecture supports two prominent training paradigms as described in Sec 3: (1) **Regression** trained with L2 loss for direct action prediction, and (2) a **Flow-Matching method** that predicts the target vector field for generative modeling.

Regarding the training setup, unless otherwise specified, we follow the hyperparameters listed in Table 2 to **train all models**. All training procedures are conducted using 8 NVIDIA A100 GPUs, **resulting in an overall computational cost exceeding 16,000 GPU-hours.**

| Configuration | Value |
|---|---|
| Optimizer | AdamW |
| Batch size | 512 |
| Learning rate | $1 \times 10^{-4}$ |
| LR Scheduler | CosineAnnealingLR |
| Weight decay | 0.01 |
| Optimizer momentum | $\beta_1, \beta_2 = 0.9, 0.95$ |
| Model precision | float32 |
| Image Resize | 224x224 |
| Image Augmentation | ColorJitter(0.2, 0.2, 0.2, 0) |

*Table 2.* Hyperparameters for model training.

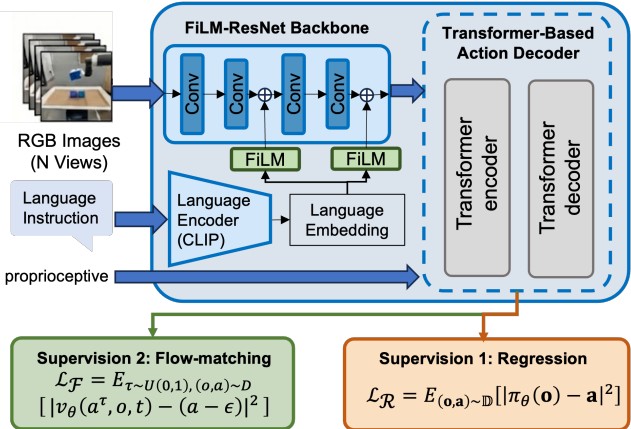

*Figure 7.* Overview of Model architecture.

## E. Details of Experimental Setup

### E.1. Real-World Experiments

In this section, we provide a detailed description of the hardware configurations and task suite used to evaluate the performance of different action spaces in real-world settings. Our empirical study is conducted on three distinct hardware platforms across four tasks, as summarized in Table 3. Visualizations of the hardware setups are provided in Fig. 8.

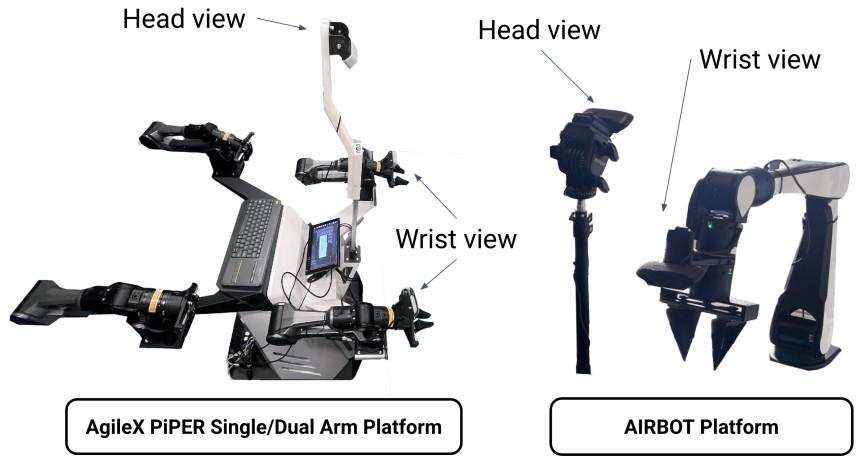

*Figure 8.* Overview of hardware setup for real world experiments

*Table 3.* **Task Curriculum, Complexity Progression, and Success Criteria.** We design four tasks with increasing difficulty to stress different failure modes of action representations and define rigorous evaluation metrics for each.

| Task | Goal | Progress Score | Evaluation Focus |
|---|---|---|---|
| **Touch Cube** | **Sanity Check**: Move from a random pose to touch a fixed cube. | 0.5 for aiming the correct direction; 0.75 for touching the cube corner; 1.0 for touching the top surface. | Isolates *spatial precision* from dynamics; validates basic convergence. |
| **Pick Cup** | **Grasping**: Grasp and lift a target cup to a height of 10 cm. | 0.5 for touching the cup; 1.0 for picking up the cup. | Introduces *contact dynamics* and gripper timing coordination. |
| **Pick & Place** | **Sequential**: Grasp a cup and place it onto a target plate. | 0.5 for touching the cup; 0.75 for picking up the cup but failing to put it on the plate; 1.00 for successfully placing it on the plate. | Sensitive to *temporal drift* and error accumulation over a long horizon. |
| **Bimanual Transfer** | **Coordination**: Transfer a cube from the left arm to a bowl in the right arm. | 0.25 for touching either the cube or the bowl; 0.5 for pick up either; 0.75 for picking up both but failing to put the cube into the bowl; 1.0 for aiming correctly. | Requires *inter-arm coordination* and precise relative positioning. |

**Single-Arm AgileX PiPER:** The single-arm PIPER is a lightweight 6-DoF robotic manipulator, which we use as a baseline platform for single-handed manipulation tasks. The setup is equipped with a third-person camera and a wrist-mounted camera to capture global visual context and fine-grained local information, respectively. This platform evaluates the model's ability to execute precise and dynamic actions, including *Touch Cube*, *Pick Up Cup*, and *Pick and Place Cup*, without the added complexity of bimanual coordination.

**Dual-Arm AgileX PiPER Platform:** Built upon the AgileX robotics platform, this setup is designed to assess fine-grained control in a bimanual manipulation setting. In addition to a third-person camera, each arm is equipped with a wrist-mounted camera, enabling active perception and coordinated two-arm behaviors. This platform targets contact-rich scenarios that require accurate timing and cross-arm coordination, exemplified by the *Bimanual Transfer* task.

**AIRBOT:** AIRBOT is a 6-DoF robotic arm characterized by its cost-effectiveness and kinematic structure that differs from the PIPER arms. We include this platform to evaluate the cross-morphology generalization capability of different action spaces on the *Touch Cube* task.

### E.2. Simulations

We use RoboTwin-2.0 as the simulation platform to evaluate the performance of different action spaces under an idealized hardware configuration. Specifically, we adopt the AgileX embodiment provided by RoboTwin-2.0 and select 10 representa-

tive manipulation tasks: *Adjust Bottle*, *Dump Bin Bigbin*, *Grab Roller*, *Lift Pot*, *Move Playingcard Away*, *Open Laptop*, *Place Burger Fries*, *Place Container Plate*, *Press Stapler*, and *Shake Bottle Horizontally*. **The task set is chosen based on preliminary experiments, excluding tasks that are either saturated or excessively difficult, in order to provide a balanced evaluation across varying levels of complexity.** The source of expert demonstrations is the official data generation tool provided by RoboTwin-2.0. The process is fully transparent and reproducible.

### E.3. Transfer Learning with $\pi_0$

To evaluate transfer capabilities, we fine-tune the pre-trained $\pi_0$ policy using Low-Rank Adaptation (LoRA) utilizing the official codebase. The experimental suite includes three tasks: *Touch Cube*, *Pick Cup*, and *Pick & Place*. Demonstrations from all tasks are merged into a unified dataset and sampled jointly during training, performing a multi-task learning paradigm. We conduct experiments across all four combinations of spatial and temporal abstractions to analyze their distinct effects. Training is performed for 30,000 steps with a batch size of 32, with checkpoints saved and validated every 10,000 steps. All other hyperparameters align with the settings described in Table 2.

## F. Cross Validation

We conduct extensive experiments to substantiate the claims made in this work. While the main text reports the most representative results, this section presents a more comprehensive set of additional experimental evidence to cross-validate our findings and provide further support for our conclusions.

### F.1. Cross-Validation of Chunk-Wise vs. Step-Wise Delta Actions

In addition to the regression-based results reported in Sec. 4.1, we further evaluate the *step-wise delta* interface on the *Cube*, *Cup*, and *Pick and Place* tasks, and compare its performance against the *chunk-wise delta* interface using a flow-matching-based backbone. As shown in Fig. 9, both chunk-wise delta end-effector and joint-space representations consistently outperform their step-wise delta counterparts.

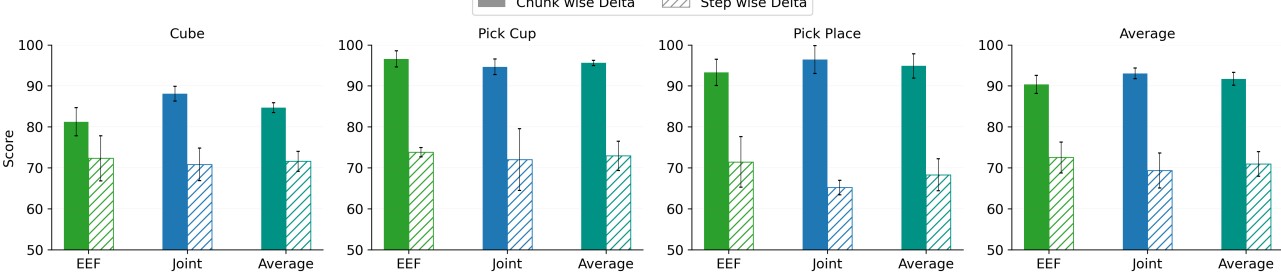

*Figure 9.* Cross-validation with a flow-matching backbone. Chunk-wise delta representations, in both end-effector and joint space, consistently outperform step-wise delta action interfaces.

### F.2. Simulation Validation: Consistency across Data and Compute Scaling

To verify that our observations regarding action abstractions are not artifacts of specific real-world hardware setups, we conducted large-scale consistency checks in simulation using the RoboTwin 2.0 benchmark. Simulation allows us to rigorously test the scaling laws of our method with significantly larger datasets and more extensive training horizons than are feasible in physical experiments.

As illustrated in Fig. 10, we observe trends that are highly consistent with our real-world findings:

- **Temporal Abstraction:** *delta* actions consistently dominate *absolute* actions across all data volumes and training epochs, confirming that relative motion control provides a more stable learning signal regardless of the regime.

- **Spatial Abstraction:** Similar to the real-world results, *joint-space* representations exhibit superior scaling properties. While Task-space control remains competitive in low-data regimes, Joint-space performance improves significantly as data volume increases, eventually outperforming Task-space by a clear margin.

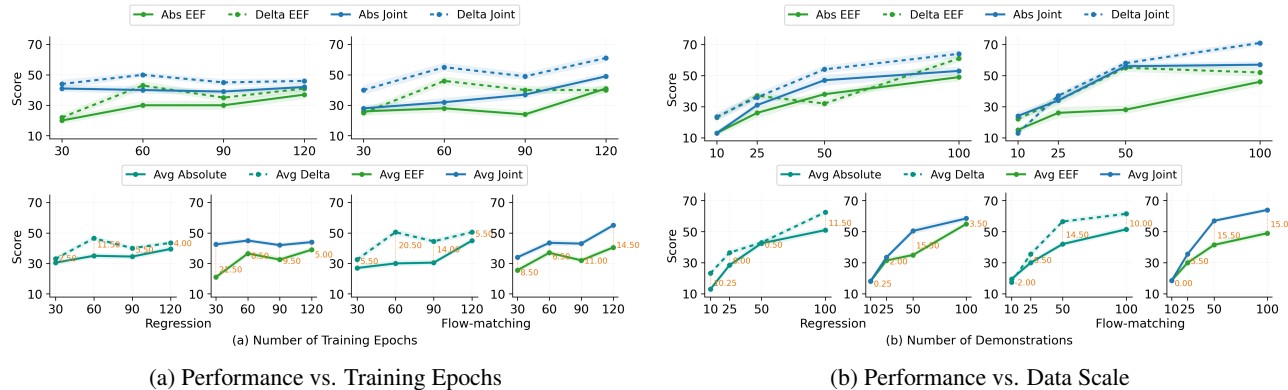

*Figure 10.* **Results on RoboTwin 2.0 cross-validate the scaling laws of action abstractions.** Comparison of policy performance across (a) training epochs and (b) demonstration scales for regression-based and flow-matching-based backbones. These results confirm that the proposed method scales effectively with both increased compute and data.

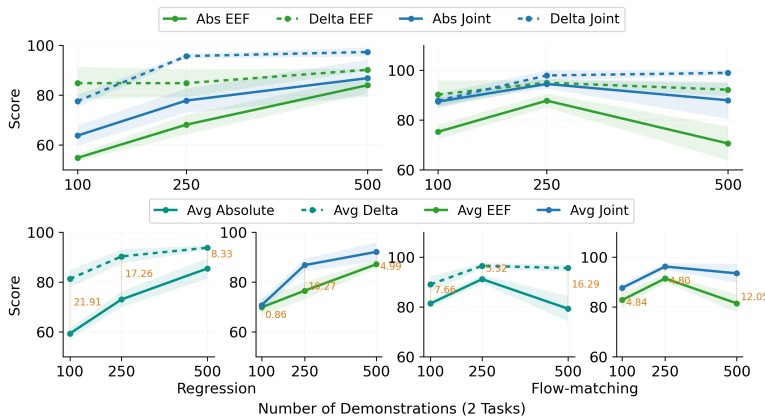

*Figure 11.* Scaling experiments on multi-task learning

These simulation results provide strong evidence that our conclusions on action space selection are fundamental to the learning dynamics of embodied agents, rather than being specific to a particular robot morphology or physical environment.

### F.3. Cross-Validation in Multi-Task Settings

Beyond single-task mastery, we further validate our approach in a multi-task learning (MTL) setting to ensure that our findings are robust to task interference and varying task distributions. We train a single, unified policy co-conditioned on multiple distinct manipulation tasks to assess how well different action abstractions handle the increased complexity across different data volumes and training epochs. The results, reported in Fig. 11, indicate that the established trends remain robust.

# G. Formal Definition and Discussion on Action Space Design

We address the problem of imitation learning for robotic manipulation, formally modeled as learning a policy $\pi_\theta$ that maps observations $\mathbf{o}_t$ to low-level deployable joint commands $\mathbf{u}_t \in \mathbb{R}^{d_q}$ at timestep $t$.

The policy produces a latent sequence $\mathbf{Z}_t \in \mathbb{R}^{c \times d_a}$, which is subsequently transformed through a two-stage process: *temporal decoding* into an executable sequence $\tilde{\mathbf{A}}_t \in \mathbb{R}^{c \times d_a}$, followed by *spatial projection* into the robot's execution space. An overview of this pipeline is provided in Fig. 12. Here, $d_q$ and $d_a$ denote the dimensions of joint-space commands and action representations, respectively, and $c \in \mathbb{N}$ denotes the chunk length. Prior work has both theoretically and empirically demonstrated that $c > 1$ is critical for effective robotic learning.

## G.1. Formalization of Action Space Design

**Temporal Decoding**   Let $\mathbf{Z}_t = [\mathbf{z}_{t,1}, \ldots, \mathbf{z}_{t,c}]^\top \in \mathbb{R}^{c \times d_a}$ denote the sequence of latent codes. These latents are decoded into an action sequence $\tilde{\mathbf{A}}_t$ using either a *zeroth-order* (absolute) or *first-order* (incremental) parameterization. In the zeroth-order case, each latent directly specifies the corresponding action. In the first-order case, actions are defined relative to a reference state $\mathbf{s}_t^{\mathrm{ref}} \in \mathbb{R}^{d_a}$:

$$\tilde{\mathbf{a}}_{t+k} = \begin{cases} \mathbf{s}_t^{\mathrm{ref}} + \mathbf{z}_{t,k}, & \text{Chunk}, \\ \mathbf{s}_t^{\mathrm{ref}} + \sum_{j=1}^{k} \mathbf{z}_{t,j}, & \text{Step}. \end{cases} \tag{1}$$

This decoding induces a linear temporal operator $\mathbf{M}_{\mathrm{time}}$ acting on the stacked latent sequence, whose structure is determined by the choice of temporal parameterization.

**Spatial Mapping**   Each decoded action $\tilde{\mathbf{a}}_{t+k}$ is projected into joint space via a spatial operator that optionally depends on the current joint configuration $\mathbf{q}_t \in \mathbb{R}^{d_q}$. We define $\Phi_{\mathrm{IK}} : \mathbb{R}^{d_a} \times \mathbb{R}^{d_q} \to \mathbb{R}^{d_q}$, which implements inverse kinematics mapping task-space targets to joint-space commands. The resulting joint command is

$$\mathbf{u}_{t+k} = \begin{cases} \tilde{\mathbf{a}}_{t+k}, & \text{joint-space}, \\ \Phi_{\mathrm{IK}}(\tilde{\mathbf{a}}_{t+k}, \mathbf{q}_t), & \text{task-space}. \end{cases} \tag{2}$$

This formulation allows the same temporal decoder to pair naturally with either joint or task-space action parameterizations, with $\Phi_{\mathrm{IK}}$ providing the necessary projection into executable joint commands.

**Combined Linear Approximation and Structural Instability**
To understand how temporal and spatial parameterizations jointly affect execution stability, we analyze the local sensitivity of the full action transformation. Small perturbations in the latent sequence should not produce disproportionately large deviations in the executed commands. To make this dependence explicit, we linearize the composite mapping from latent codes $\mathbf{Z}_t$ to joint-space actions. Let $\mathcal{T}_{\mathrm{space}}$ denote the spatial projection and define its local Jacobian

$$\mathbf{S}_t = \left. \frac{\partial \mathcal{T}_{\mathrm{space}}}{\partial \tilde{\mathbf{a}}} \right|_{\tilde{\mathbf{a}}=\tilde{\mathbf{a}}_t} \in \mathbb{R}^{d_q \times d_a}. \tag{3}$$

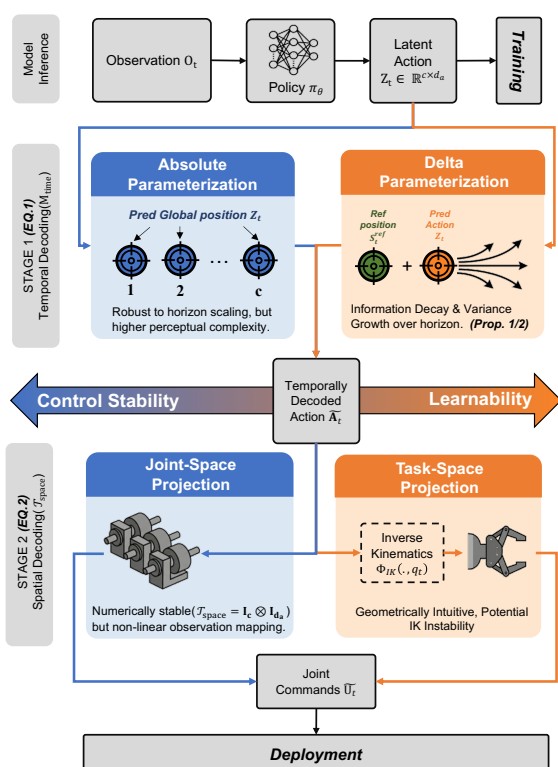

*Figure 12.* Problem reformulation.

For joint-space parameterizations $\mathbf{S}_t = \mathbf{I}_{d_q}$; for task-space mappings, $\mathbf{S}_t$ may represent a differential IK Jacobian or any differentiable projection. Stacking across the horizon yields the block-diagonal operator $\mathbf{S}_{\mathrm{space}} = \mathbf{I}_k \otimes \mathbf{S}_t$. The resulting first-order approximation of the full transformation is

$$\mathcal{T}_{\mathrm{total}} \approx (\mathbf{I}_k \otimes \mathbf{S}_t) \mathbf{M}_{\mathrm{time}}. \tag{4}$$

This expression shows that temporal and spatial representations contribute multiplicatively to overall stability: the sensitivity is governed jointly by the spectral properties of the spatial factor $\mathbf{S}_t$ and the temporal operator $\mathbf{M}_{\text{time}}$.

### G.2. Research Question on Temporal Reparameterization

Temporal reparameterization determines how errors accumulate over the horizon, how rapidly information decays, and how difficult the resulting mapping is for a policy to learn from a single observation. To understand these trade-offs, we analyze the behavior of different temporal parameterizations below. Here, we recall the Proposition 4.1, and given the detailed proof.

**Proposition G.1** (Noise Amplification in Step-wise Integration). *The step-wise delta representation corresponds to the linear temporal operator* $\mathbf{M}_{\text{step}} = \mathbf{L}_k \otimes \mathbf{I}_{d_a}$, *where* $\mathbf{L}_k \in \mathbb{R}^{k \times k}$ *is the lower-triangular cumulative-sum matrix (with ones on and below the diagonal). The spectral norm of this operator satisfies:*

$$\|\mathbf{M}_{\text{step}}\|_2 = \sigma_{\max}(\mathbf{L}_k) \approx \frac{2k+1}{\pi}.$$

*This norm grows linearly with $c$ and strictly exceeds $1$ for all $c \geq 2$.*

Consequently, step-wise integration necessarily amplifies input prediction noise as the horizon length increases, inducing structural instability.

*Proof.* We seek to bound the spectral norm $\|\mathbf{M}_{\text{step}}\|_2$, where $\mathbf{M}_{\text{step}} = \mathbf{L}_k$ is the $k \times k$ lower-triangular matrix of ones. Recall that the spectral norm is given by the largest singular value: $\|\mathbf{L}_k\|_2 = \sigma_{\max}(\mathbf{L}_k)$.

Directly computing the eigenvalues of $\mathbf{L}_k \mathbf{L}_k^T$ is complex. Instead, we analyze the inverse operator. The inverse of the cumulative sum matrix is the discrete difference operator, denoted as $\mathbf{D}_k = \mathbf{L}_k^{-1}$:

$$\mathbf{D}_k = \begin{pmatrix} 1 & 0 & \cdots & 0 \\ -1 & 1 & \cdots & 0 \\ \vdots & \ddots & \ddots & \vdots \\ 0 & \cdots & -1 & 1 \end{pmatrix}.$$

Using the property of singular values, $\sigma_{\max}(\mathbf{L}_k) = \frac{1}{\sigma_{\min}(\mathbf{D}_k)}$. We compute the eigenvalues of the symmetric matrix $\mathbf{A} = \mathbf{D}_k \mathbf{D}_k^T$. The matrix $\mathbf{A}$ takes the form of a tridiagonal matrix (closely related to the discrete Laplacian):

$$\mathbf{A} = \begin{pmatrix} 1 & -1 & 0 & \cdots \\ -1 & 2 & -1 & \cdots \\ \vdots & \ddots & \ddots & \vdots \\ 0 & \cdots & -1 & 2 \end{pmatrix}.$$

The eigenvalues $\lambda_i$ of this specific tridiagonal matrix are well-known in numerical analysis (Smith, 1985):

$$\lambda_i = 2 - 2\cos\left(\frac{(2i-1)\pi}{2k+1}\right) = 4\sin^2\left(\frac{(2i-1)\pi}{2(2k+1)}\right), \quad i = 1, \ldots, k.$$

The singular values of $\mathbf{D}_k$ are the square roots of these eigenvalues: $\sigma_i(\mathbf{D}_k) = 2\sin\left(\frac{(2i-1)\pi}{2(2k+1)}\right)$. The minimum singular value corresponds to $i = 1$:

$$\sigma_{\min}(\mathbf{D}_k) = 2\sin\left(\frac{\pi}{2(2k+1)}\right).$$

Therefore, the spectral norm of the cumulative sum matrix is:

$$\|\mathbf{L}_k\|_2 = \frac{1}{\sigma_{\min}(\mathbf{D}_k)} = \frac{1}{2\sin\left(\frac{\pi}{4k+2}\right)}.$$

Using the Taylor expansion $\sin(x) \approx x$ for small $x$ (valid as $k$ grows):

$$\|\mathbf{L}_k\|_2 \approx \frac{1}{2 \cdot \frac{\pi}{4k+2}} = \frac{2k+1}{\pi}.$$

This confirms that the error amplification factor grows linearly with the chunk size $k$, proving the proposition. □

**Remark on Temporal Decorrelation for Long-Horizon Action**    Although the Chunk-Delta and Absolute formulation $\tilde{\mathbf{a}}_{t+k} = \mathbf{s}_t + \mathbf{z}_{t,k}$ avoids numerical drift from cumulative integration, it remains fundamentally *open-loop*: each offset $\mathbf{z}_{t,k}$ must be predicted from the single observation $\mathbf{o}_t$ without intermediate feedback. Let the regression or generative target be the displacement $\Delta \mathbf{a}_k^* = \mathbf{a}_{t+k}^* - \mathbf{s}_t$. Any model can only approximate its conditional distribution $p(\Delta \mathbf{a}_k^* \mid \mathbf{o}_t)$, whose inherent uncertainty is characterized by the conditional entropy $H(\Delta \mathbf{a}_k^* \mid \mathbf{o}_t)$. As the prediction horizon increases, two mechanisms enlarge this uncertainty:

1. **Variance Growth.** The displacement magnitude $\|\Delta \mathbf{a}_k^*\|$ generally increases with $k$, expanding the support of the target distribution and increasing its entropy.

2. **Information Decay.** The mutual information $I(\mathbf{a}_{t+k}^*; \mathbf{o}_t)$ decreases with $k$, reflecting the diminishing predictive power of $\mathbf{o}_t$. Consequently, the conditional entropy $H(\Delta \mathbf{a}_k^* \mid \mathbf{o}_t)$ must increase.

**Remark on Absolute Stability but Increased Learning Difficulty**    In contrast to Delta representations, Absolute parameterization is statistically robust to horizon scaling: the regression targets $\mathbf{a}_{t+k}$ remain confined to a bounded workspace, preventing variance growth. Moreover, Absolute predictions map observations directly to global task-space coordinates, avoiding the stale-reference effect inherent in relative corrections. However, this robustness comes at the cost of substantially increased learning difficulty. Because the policy must implicitly infer full-scene geometry, global localization, and workspace-scale structure from raw observations, Absolute parameterization demands a more complex perceptual model and is often harder to train effectively than Delta-based variants.

### G.3. Research Question on Spatial Reparameterization

Beyond temporal structure, the choice of spatial action manifold also imposes strong inductive biases on both the learnability and stability of the control policy. Whereas temporal parameterization governs how predictions evolve over the horizon, spatial parameterization determines the geometry of the control interface itself and thus fundamentally affects the numerical conditioning and perceptual complexity of the policy. **Task-space** control offers a geometrically meaningful abstraction but introduces the potentially ill-conditioned pseudo-inverse Jacobian $\mathbf{J}^{\dagger}(\mathbf{q}_t)$ into $\mathcal{T}_{\text{total}}$. In contrast, **Joint-space** control is numerically stable by construction, yet forces the policy to regress visual observations into a highly non-linear configuration space, significantly increasing perceptual complexity.

### G.4. Summary

Together, these analyses reveal that action parameterization fundamentally governs a core trade-off between **learnability** (the functional complexity of the mapping $f : \mathbf{o}_t \rightarrow \mathbf{Z}_t$) and **stability** (the numerical conditioning of the final control transformation $\mathcal{T}_{\text{total}}$). We structure our empirical study around two orthogonal axes: temporal structure and spatial manifold, aiming to identify action parameterizations that most effectively facilitate robust visuomotor policy learning.

## H. Detailed Statistics

In this section, we provide all detailed statistics of our reported results.

*Table 4.* Task success rates under different data regimes

| Method | Control | # Data | Cube | Pick Cup | Pick and Place | Bimanual Transfer | Average |
|--------|---------|--------|------|----------|----------------|-------------------|---------|
| DP | abs-ee | 100 | 58.70 | 60.87 | 69.57 | 56.52 | 61.41 |
| | rel-ee | | 80.43 | 84.78 | 69.57 | 45.65 | 70.11 |
| | abs-joint | | 78.26 | 60.87 | 60.87 | 57.61 | 64.40 |
| | rel-joint | | 86.96 | 89.13 | 73.91 | 58.70 | 77.17 |
| | abs-ee | 250 | 83.70 | 65.22 | 73.91 | 65.22 | 72.01 |
| | rel-ee | | 91.30 | 97.83 | 84.78 | 76.09 | 87.50 |
| | abs-joint | | 95.65 | 78.26 | 81.52 | 75.00 | 82.61 |
| | rel-joint | | 96.74 | 97.83 | 93.48 | 75.00 | 90.76 |
| | abs-ee | 500 | 93.48 | 73.91 | 76.09 | 78.26 | 80.43 |
| | rel-ee | | 86.96 | 100.00 | 97.83 | 71.74 | 89.13 |
| | abs-joint | | 100.00 | 65.22 | 85.87 | 85.65 | 84.18 |
| | rel-joint | | 100.00 | 97.83 | 91.30 | 84.78 | 93.48 |
| ACT | abs-ee | 100 | 76.09 | 54.35 | 57.61 | 59.78 | 61.96 |
| | rel-ee | | 93.48 | 80.56 | 70.65 | 57.61 | 75.57 |
| | abs-joint | | 70.65 | 75.00 | 55.43 | 43.48 | 61.14 |
| | rel-joint | | 91.30 | 77.78 | 67.39 | 42.39 | 69.72 |
| | abs-ee | 250 | 77.17 | 63.04 | 66.30 | 63.04 | 67.39 |
| | rel-ee | | 85.87 | 97.83 | 84.78 | 71.70 | 85.04 |
| | abs-joint | | 77.17 | 84.78 | 70.65 | 52.17 | 71.20 |
| | rel-joint | | 84.78 | 95.65 | 83.70 | 69.57 | 83.42 |
| | abs-ee | 500 | 96.74 | 76.09 | 56.52 | 68.48 | 74.46 |
| | rel-ee | | 97.83 | 100.00 | 96.74 | 65.22 | 89.95 |
| | abs-joint | | 98.91 | 95.65 | 77.17 | 72.83 | 86.14 |
| | rel-joint | | 100.00 | 100.00 | 89.13 | 76.09 | 91.30 |

*Table 5.* Task success rates under different training epochs

| Method | Control | # Epochs | Cube | Pick Cup | Pick and Place | Bimanual Transfer | Average |
|---|---|---|---|---|---|---|---|
| ACT | abs-ee | 300 | 66.30 | 52.38 | 66.30 | 61.96 | 61.74 |
| | rel-ee | | 73.91 | 92.86 | 91.30 | 67.39 | 81.37 |
| | abs-joint | | 65.22 | 64.29 | 67.39 | 51.09 | 62.00 |
| | rel-joint | | 82.61 | 95.24 | 92.39 | 64.13 | 83.59 |
| | abs-ee | 600 | 77.17 | 63.04 | 66.30 | 63.04 | 67.39 |
| | rel-ee | | 85.87 | 97.83 | 84.78 | 67.39 | 83.97 |
| | abs-joint | | 77.17 | 84.78 | 70.65 | 52.17 | 71.20 |
| | rel-joint | | 84.78 | 95.65 | 83.70 | 69.57 | 83.42 |
| | abs-ee | 900 | 80.43 | 65.22 | 58.70 | 71.74 | 69.02 |
| | rel-ee | | 85.87 | 92.39 | 82.61 | 66.30 | 81.79 |
| | abs-joint | | 83.70 | 73.91 | 61.96 | 58.70 | 69.57 |
| | rel-joint | | 82.61 | 95.65 | 89.13 | 68.48 | 83.97 |
| | abs-ee | 1200 | 83.70 | 71.74 | 68.48 | 72.83 | 74.18 |
| | rel-ee | | 86.96 | 97.83 | 89.13 | 71.74 | 86.41 |
| | abs-joint | | 86.96 | 86.96 | 68.48 | 66.30 | 77.17 |
| | rel-joint | | 84.78 | 100.00 | 95.65 | 67.39 | 86.96 |
| DP | abs-ee | 300 | 79.35 | 58.70 | 65.22 | 65.22 | 67.12 |
| | rel-ee | | 93.48 | 100.00 | 82.61 | 75.00 | 87.77 |
| | abs-joint | | 90.22 | 65.22 | 67.39 | 63.04 | 71.47 |
| | rel-joint | | 96.74 | 95.65 | 91.30 | 67.39 | 87.77 |
| | abs-ee | 600 | 83.70 | 65.22 | 73.91 | 65.22 | 72.01 |
| | rel-ee | | 91.30 | 97.83 | 84.78 | 76.09 | 87.50 |
| | abs-joint | | 95.65 | 78.26 | 81.52 | 75.00 | 82.61 |
| | rel-joint | | 96.74 | 97.83 | 93.48 | 75.00 | 90.76 |
| | abs-ee | 900 | 66.30 | 67.39 | 80.43 | 81.52 | 73.91 |
| | rel-ee | | 88.04 | 89.13 | 82.61 | 70.65 | 82.61 |
| | abs-joint | | 88.04 | 73.91 | 75.00 | 73.91 | 77.72 |
| | rel-joint | | 95.65 | 97.83 | 86.96 | 78.26 | 89.67 |
| | abs-ee | 1200 | 80.43 | 69.57 | 78.26 | 79.35 | 76.90 |
| | rel-ee | | 95.65 | 95.65 | 92.39 | 71.74 | 88.86 |
| | abs-joint | | 93.48 | 76.09 | 76.09 | 73.91 | 79.89 |
| | rel-joint | | 97.83 | 100.00 | 100.00 | 79.35 | 94.29 |

*Table 6.* Task performance of ACT and DP averaged over three independent trials.

| Task | Method | abs-ee | rel-ee | abs-qpos | rel-qpos |
|---|---|---|---|---|---|
| Adjust Bottle | ACT | $36.7 \pm 3.3$ | $33.3 \pm 6.7$ | $23.3 \pm 3.3$ | $40.0 \pm 0.0$ |
| | DP | $40.0 \pm 11.5$ | $30.0 \pm 5.8$ | $20.0 \pm 5.8$ | $60.0 \pm 5.8$ |
| Dump Bin Bigbin | ACT | $26.7 \pm 13.3$ | $50.0 \pm 20.0$ | $50.0 \pm 5.8$ | $43.3 \pm 6.7$ |
| | DP | $16.7 \pm 6.7$ | $53.3 \pm 8.8$ | $56.7 \pm 13.3$ | $53.3 \pm 6.7$ |
| Grab Roller | ACT | $23.3 \pm 14.5$ | $40.0 \pm 10.0$ | $56.7 \pm 14.5$ | $93.3 \pm 3.3$ |
| | DP | $30.0 \pm 15.3$ | $66.7 \pm 18.6$ | $33.3 \pm 12.0$ | $80.0 \pm 5.8$ |
| Lift Pot | ACT | $20.0 \pm 20.0$ | $3.3 \pm 3.3$ | $33.3 \pm 6.7$ | $23.3 \pm 8.8$ |
| | DP | $3.3 \pm 3.3$ | $13.3 \pm 8.8$ | $13.3 \pm 3.3$ | $13.3 \pm 13.3$ |
| Move Playingcard Away | ACT | $3.3 \pm 3.3$ | $16.7 \pm 12.0$ | $23.3 \pm 14.5$ | $36.7 \pm 3.3$ |
| | DP | $10.0 \pm 5.8$ | $26.7 \pm 3.3$ | $16.7 \pm 8.8$ | $43.3 \pm 14.5$ |
| Open Laptop | ACT | $10.0 \pm 0.0$ | $30.0 \pm 10.0$ | $30.0 \pm 5.8$ | $23.3 \pm 8.8$ |
| | DP | $13.3 \pm 3.3$ | $16.7 \pm 8.8$ | $13.3 \pm 6.7$ | $16.7 \pm 8.8$ |
| Place Burger Fries | ACT | $13.3 \pm 3.3$ | $20.0 \pm 10.0$ | $30.0 \pm 5.8$ | $46.7 \pm 6.7$ |
| | DP | $13.3 \pm 3.3$ | $16.7 \pm 6.7$ | $23.3 \pm 3.3$ | $30.0 \pm 10.0$ |
| Place Container Plate | ACT | $26.7 \pm 3.3$ | $40.0 \pm 10.0$ | $53.3 \pm 6.7$ | $60.0 \pm 5.8$ |
| | DP | $26.7 \pm 6.7$ | $30.0 \pm 5.8$ | $50.0 \pm 5.8$ | $50.0 \pm 10.0$ |
| Press Stapler | ACT | $13.3 \pm 3.3$ | $13.3 \pm 3.3$ | $23.3 \pm 8.8$ | $30.0 \pm 10.0$ |
| | DP | $16.7 \pm 3.3$ | $23.3 \pm 3.3$ | $13.3 \pm 3.3$ | $30.0 \pm 10.0$ |
| Shake Bottle Horiz. | ACT | $93.3 \pm 6.7$ | $86.7 \pm 3.3$ | $76.7 \pm 12.0$ | $66.7 \pm 3.3$ |
| | DP | $90.0 \pm 5.8$ | $93.3 \pm 6.7$ | $83.3 \pm 12.0$ | $96.7 \pm 3.3$ |
| **Avg (Overall)** | **ACT** | $\mathbf{26.7 \pm 3.3}$ | $\mathbf{33.3 \pm 6.1}$ | $\mathbf{40.0 \pm 0.6}$ | $\mathbf{46.3 \pm 1.9}$ |
| | **DP** | $\mathbf{26.0 \pm 1.2}$ | $\mathbf{37.0 \pm 6.2}$ | $\mathbf{32.3 \pm 2.6}$ | $\mathbf{48.0 \pm 4.4}$ |

