# OpenReview forum: "Demystifying Action Space Design for Robotic Manipulation Policies"
_ICML.cc/2026/Conference — ICML 2026 regular_

### Official Review · Reviewer_UfzG · 2026-02-20

**Soundness:** 3
**Presentation:** 3
**Significance:** 3
**Originality:** 3
**Overall Recommendation:** 4
**Confidence:** 4

**Summary:**

This paper conducts a large-scale empirical study on action space design for imitation learning in robotic manipulation. It analyzes temporal (absolute vs. delta) and spatial (joint vs. task space) action abstractions across multiple tasks and platforms. The results show that delta actions consistently outperform absolute actions, while joint-space control excels in in-domain performance and task-space representations benefit cross-embodiment transfer. The paper provides practical guidelines in robot action design of manipulation tasks.

**Compliance With Llm Reviewing Policy:**

Affirmed.

**Final Justification:**

I raise the score from 3 to 4. Overall, it is a very comprehensive study on action space design for VLA models with solid experiments and results. The authors provides further experiments and justifications which addressed my concerns.

**Key Questions For Authors:**

My key questions are:
1. Will the observed trends hold in more spatially complex, highly dynamic, or dexterous settings?
2. Actually, apart from Cartesian/ Joint Space commands, there are other paradigms like object-centric/hierarchical action space design ([1], [2], [3]). What do you think about their role in action space design? Including discussion about them in the paper will make it stronger.

I will raise the score if the authors address the questions in the rebuttal.

[1] Zhou, Wenxuan, et al. "HACMan: Learning Hybrid Actor-Critic Maps for 6D Non-Prehensile Manipulation." 7th Annual Conference on Robot Learning, 2023.

[2] Jiang, Bowen, et al. "HACMan++: Spatially-Grounded Motion Primitives for Manipulation." Robotics: Science and Systems, 2024.

[3] Xue, Han, et al. "Reactive Diffusion Policy: Slow-Fast Visual-Tactile Policy Learning for Contact-Rich Manipulation." Proceedings of Robotics: Science and Systems (RSS), 2025.

**Limitations:**

Yes.

**Strengths And Weaknesses:**

## Strengths

- **Comprehensive empirical evaluation:** The paper presents a large-scale real-world study, providing strong empirical support for its conclusions.

- **Systematic and well-structured framework:** The decomposition of action space into temporal and spatial axes offers a clear and organized way to analyze design choices.

- **Actionable practical insights:** The study provides concrete and implementable recommendations (e.g., chunk-wise delta, abstraction-dependent horizon tuning) that are directly useful for practitioners.

- **Robust validation across diverse settings:** Experiments span multiple platforms, model variants, and transfer/cross-embodiment regimes, increasing confidence in the reported trends.

## Weaknesses

- **Limited conceptual novelty:** The paper studies important design axes (absolute vs. delta, joint vs. task space), but these dimensions have been explored seperately in prior work. The main contribution is large-scale empirical consolidation rather than introducing novel action space design methodology.

- **Task complexity scope:** The evaluation focuses on relatively structured manipulation tasks. It remains unclear whether the observed trends would hold in more spatially complex, highly dynamic, or dexterous settings (e.g., push-T, cloth folding, in-hand manipulation) that require stronger spatial reasoning and fine-grained control.

- **Limited coverage of action design:** The paper did not cover hierarchical/ object-centric views of action space design in manipulation, and is limited to Cartesian/ Joint space.

---

> ### Author Rebuttal · Authors · 2026-03-31
>
> Thank you for the thoughtful review and for recognizing the empirical breadth and practical value of our study. We respond to each point below:
>
> > W1: Limited Conceptual Novelty
>
> We agree that this is not a methodology paper proposing a new model. Instead, as highlighted in the main text, our core contribution is providing the **first comprehensive, large-scale, controlled empirical investigation** to demystify action space design for robot manipulation. The current community lacks a systematic understanding of how different action spaces impact different tasks and training regimes. Our work fills this gap, turning fragmented observations into actionable, robust guidelines for the community.
>
> > W2/Q1: Task complexity scope
>
> We appreciate this suggestion and have added new experiments on cloth folding, a highly spatially complex and deformable manipulation task. We specifically selected [Soft-Fold](https://huggingface.co/datasets/Facebear/XVLA-Soft-Fold), an open-source cloth folding dataset that ensures our new benchmarks remain fully reproducible. To comprehensively evaluate the action spaces within this domain, we designed two difficulty levels: folding a cloth from a fixed, flattened state (Easy) and from a random state (Hard). We trained a VLM+DiT-based VLA model and utilizing X-VLA [1] weights for the pretrained setting. We report success rates over 5 independent trials for the easy setting and 3 trials for the hard setting.
>
>
> |          Setting         | Delta     EEF     |       Delta Joint          | Absolute  EEF    |    Absolute Joint    |
> | ------------------ | --------------- | --------------- | ------------- | ----- |
> | w/o Pretrain  (Easy) | 2/5            | **3/5** | 1/5          | *2/5*  |
> | w/o Pretrain  (Hard) | 0/3             | **1/3**   | 0/3           | 0/3   |
> | w Pretrain (Easy) | **5/5** | **5/5** | *4/5*          | *4/5*  |
> | w Pretrain (Hard) | **3/3**   | **2/3**   | **2/3** | *2/3*   |
>
> These results closely align with the key takeaways in our main text:
>
> 1. Delta actions consistently outperform absolute actions and joint-space control generally provide a more robust action representation for poliy learning compared to EEF-space.
> 2. Large-scale pretraining significantly benefits the EEF-based action space. With X-VLA pretraining, both Delta EEF and Absolute EEF see massive improvements, matching or surpassing the best Joint-space performance.
>
> This complex setup provides additional evidence supporting the credibility and generalizability of our conclusions. We will incorporate these results and analyses into the revised main text. We also provide extra demos and analysis [here](https://sites.google.com/view/icml-2026-rebuttal).
>
>
> > W3/Q2: Limited coverage of action design
>
> We fully agree that object-centric, hierarchical, and learning-based action spaces are highly promising directions; indeed, we discuss several of these advanced formulations in our Appendix. However, we would like to clarify that our investigation scope is driven by the current landscape of large-scale embodied datasets (e.g., Open X-Embodiment [2], RoboMind [3], AgiBot-beta [4]). The action representations we focus on are the most widely available and mainstream in these repositories. Understanding these fundamental spaces is an urgent priority for the community, as the scalability of current and next-generation foundation VLAs depends heavily on standardized, easily accessible action labels.
>
> In contrast, advanced action labels often require additional modalities (e.g., point clouds) or are exceedingly difficult to retroactively extract at scale from existing diverse datasets. Due to this scalability bottleneck, we excluded them from our current empirical scope. That said, the papers provided by reviewer are excellent references for the future of action abstraction design; **we will incorporate a deeper discussion of these methods and how they might address the limitations identified in our study in the revised main text.**
>
> [1] X-vla: Soft-prompted transformer as scalable cross-embodiment vision-language-action model. ICLR 2025.
>
> [2] Open x-embodiment: Robotic learning datasets and rt-x models, 2024.
>
> [3] Robomind: Benchmark on multi-embodiment intelligence normative data for robot manipulation. 2025.
>
> [4] Agibot world colosseo: A large-scale manipulation platform for scalable and intelligent embodied systems. 2025.

---

> > ### Author Rebuttal · Reviewer_UfzG · 2026-04-01
> >
> > The authors provides new experiments and justifications, which addresses my concerns.

---

> > > ### Author Response · Authors · 2026-04-06
> > >
> > > We thank the reviewer for the feedback and are glad that the additional experiments and clarifications have addressed the concerns. We will incorporate these updates into the revision.

---

### Official Review · Reviewer_59CY · 2026-03-10

**Soundness:** 4
**Presentation:** 4
**Significance:** 4
**Originality:** 4
**Overall Recommendation:** 6
**Confidence:** 5

**Summary:**

This work investigates how different action‑space representations affect robot learning. Using extensive experiments in both simulated and real‑world robotic environments, the authors study two main dimensions: **spatial abstraction** (joint space vs. end‑effector space) and **temporal abstraction** (absolute actions vs. delta‑based action chunks). The goal is to derive practical guidance on which representations work best, especially given that all of these variants appear in recent state‑of‑the‑art methods.

The experiments also examine several additional factors that influence performance: **temporal horizon length**, **policy architecture** (correlation‑based vs. diffusion‑based), and **embodiment transfer**.

**Compliance With Llm Reviewing Policy:**

Affirmed.

**Final Justification:**

No need to change my original rating, this is strong work with important contributions so definitely accept.

**Key Questions For Authors:**

I have no further questions, excellent job and research paper - I hope it would be mine ;-)

**Limitations:**

yes

**Strengths And Weaknesses:**

Overall, the work is of excellent quality. It is well written and addresses important and timely questions in robot learning. The results are convincing, and the study is likely to become influential for future research in action‑space design and robot learning more broadly.

---

### Weaknesses / Points for Improvement
1. Some acronyms are not defined on first use. For example, *EEF* appears without explanation, which may confuse readers who skim figures and results.

2. The purpose of the “average” plots in Figure 3 is unclear. It would help to explain what is being averaged and why these plots are necessary.

3. Plot captions would benefit from explicitly stating whether the results come from simulated or real‑world experiments. This is clear from the main text, but adding it to captions improves readability for quick or selective readers.

4. If the RoboTwin 2.0 (simulated) results for each task in Table 1 are located in the appendix, this should be mentioned explicitly.

5. The “Progress score” metric is not clearly defined. If it corresponds to task success rate, this should be stated directly. Using the term *Success rate* in the plots might also improve clarity.

6. The source of “expert demonstrations” for simulated tasks is not explained. If these demonstrations come from a pre‑trained policy, this should be clarified.

---

> ### Author Rebuttal · Authors · 2026-03-31
>
> Thank you for the encouraging review. We are glad that you find our work well-written, timely, and likely to be influential. We address each presentation suggestion below and will incorporate all of them in the revised manuscript.
>
> > W1: Undefined acronyms (e.g., EEF)
>
> Thank you for pointing this out. We will define EEF (End-Effector) and all other acronyms on first use in the revised version.
>
> > W2: Purpose of "average" plots in Figure 3 unclear
>
> The "average" plots in Figure 3 aggregate the performance of different spatial representations (EEF / Joint) under the same delta implementation method. This averaging is designed to more directly derive Takeaway-1 (chunk-wise Implementation consistently outperforms step-wise), isolating the effect of temporal abstraction from spatial choices. We will add a clarifying note in the caption.
>
> > W3: Sim vs. real not explicit in plot captions
>
> Agreed. We will add explicit labels (Simulation / Real-World) to all figure captions for standalone readability.
>
> > W4: Reference to per-task RoboTwin-2.0 results in appendix
>
> Yes, the per-task breakdown for RoboTwin-2.0 is provided in the Appendix. We will add an explicit cross-reference in the main text where Table 1 is discussed.
>
> > W5: "Progress score" metric not clearly defined
>
> Progress score is a weighted partial-completion metric that awards credit proportional to completed sub-goals (e.g., reaching, grasping, lifting), rather than a binary success/fail. Compared to binary success rate, it provides a more fine-grained and informative signal for comparing policies. We have provided a formal definition in Appendix Table 3. We will make this more prominent and add a clear definition at its first occurrence in the main text in the revised version.
>
> > W6: Source of expert demonstrations for simulated tasks
>
> We use the official data generation tool provided by RoboTwin-2.0 to generate 50 trajectories per task. The process is fully transparent and reproducible. We will state this explicitly in the experimental setup section of the revised version.
>
> ---
>
> **Additional Update:** During the rebuttal period, we further extended our evaluation to a broader setup, including a cloth folding task, which is a highly spatially complex and deformable manipulation challenge. We provide additional demos and in-depth analysis at: https://sites.google.com/view/icml-2026-rebuttal. We hope this may be of interest.

---

> > ### Author Rebuttal · Reviewer_59CY · 2026-04-01
> >
> > Thank you for considering my minor suggestions, which I believe will help readers better appreciate your excellent paper. I have no further concerns, and I continue to find this work exceptionally strong, especially in comparison to many papers I have reviewed for top-tier ML conferences. I fully expect to make use of your findings in my own future research and will cite your work accordingly.
> >
> > I sincerely hope to see more papers of this quality in the ML community. In my view, critical comments that dismiss empirical studies or insist on introducing yet another “new model” or “method contribution” are often unnecessary. We already have an abundance of models and methods; what we lack is a deeper understanding of *how* and *why* to use them to tackle important problems. Work like yours meaningfully advances that understanding.

---

> > > ### Author Response · Authors · 2026-04-01
> > >
> > > We are glad that our response have addressed the reviewer’s concerns, and we sincerely appreciate the thoughtful comments, which have helped further improve the quality and clarity of the paper.
> > > We are also grateful for the reviewer’s encouraging feedback and perspective on the value of empirical studies. We share this view and hope that our work can contribute to a deeper understanding of existing methods and support more informed progress within the community.

---

### Official Review · Reviewer_s81P · 2026-03-11

**Soundness:** 3
**Presentation:** 3
**Significance:** 3
**Originality:** 2
**Overall Recommendation:** 5
**Confidence:** 4

**Summary:**

The paper presents a large-scale empirical study investigating how action space design impacts imitation learning for robotic manipulation. The authors decompose the action representation along two orthogonal axes: temporal (absolute vs. delta parameterization) and spatial (joint-space vs. task-space). Through over 13,000 real-world rollouts across multiple hardware platforms (single-arm AgileX, dual-arm AgileX, AIRBOT) and simulation (RoboTwin 2.0), evaluating 500+ trained models with both regression-based and flow-matching-based policies, the paper arrives at two main conclusions: (1) delta-based actions with chunk-wise alignment consistently outperform absolute actions across all settings, and (2) joint-space control is generally superior when paired with strong generative models and sufficient data, while task-space control offers advantages in cross-embodiment and transfer learning scenarios. The paper also identifies important implementation nuances, such as the superiority of chunk-wise over step-wise delta and the coupling between execution horizon and temporal abstraction choice.

**Compliance With Llm Reviewing Policy:**

Affirmed.

**Key Questions For Authors:**

1. Horizon confound in Table 1: Since delta and absolute actions use different execution horizons (k=30 vs. k=60) in the main comparison, could you provide results with a matched horizon (e.g., k=30 for both, or k=60 for both) as a reference? This would help disentangle the effect of the action representation from the effect of the horizon choice. The absence of this comparison weakens the central claim.
2. Rotation representation: What rotation parameterization was used for task-space actions? Did you experiment with alternatives (e.g., 6D rotation vs. quaternion vs. Euler)? Given that rotation representation discontinuities are a known issue in task-space control, this could be a significant confound in the spatial axis comparison.
3. IK solver sensitivity: How sensitive are the task-space results to the specific inverse kinematics solver used? Different solvers (e.g., damped least-squares vs. pseudo-inverse) handle singularities differently, which could significantly affect the joint-vs-task comparison.
4. Kinematic similarity of embodiments: How kinematically different are the AgileX PiPER and AIRBOT platforms? If they share similar workspace geometry and DoF, the cross-embodiment advantage of task-space may be overstated. Could you quantify the morphological difference?
5. Spatial variation in performance: Your grid-based protocol partitions the workspace into a 6×6 grid. Did you observe systematic spatial variation in performance (e.g., degradation near workspace boundaries or singularity regions)? This could provide additional insight into the joint-vs-task comparison.

**Limitations:**

Yes. The authors discuss limitations thoughtfully in Appendix B, including the restriction to rigid-body tasks, position-level control, and the need for validation on higher-DoF systems. The ethics statement is appropriate for the nature of this work. One additional limitation that may be worth noting: the conclusions are tied to specific policy architectures (regression and flow-matching), and may not generalize to fundamentally different paradigms (e.g., RL-based fine-tuning, world-model-based planning).

**Strengths And Weaknesses:**

Strengths:
- Scale and rigor of real-world evaluation. The sheer volume of experiments (13,000+ real-world rollouts, 2,000+ demonstrations, 500+ trained models, 16,000+ GPU-hours) is exceptional for a robotics study. This level of investment is commendable and provides improved credibility to the conclusions, setting a high bar for empirical robotics research.
Well-structured decomposition. The orthogonal decomposition into temporal and spatial axes provides a clean framework for reasoning about action spaces. The progressive research question structure (RQ1→RQ2→RQ3) builds understanding incrementally and is easy to follow.
- Practically valuable implementation insights. The chunk-wise vs. step-wise delta analysis, supported by Proposition 4.1, addresses a subtle implementation detail that practitioners frequently encounter but rarely see analyzed rigorously. The horizon-coupling finding (delta favors shorter execution windows, absolute favors longer ones) is similarly actionable.
- Cross-validation breadth. The findings are validated across simulation and real-world settings, multiple robot morphologies, both regression and flow-matching paradigms, and varying data/compute scales. The appendix provides extensive additional evidence.
Theoretical grounding. Proposition 4.1 on noise amplification in step-wise integration provides a clear and correct theoretical explanation for an empirical observation, showing the error bound grows as O(k) for step-wise but remains O(1) for chunk-wise delta.

Weaknesses:
- Inconclusive spatial axis results. While the temporal axis yields a clear and consistent recommendation (delta is better), the spatial axis conclusions are significantly more hedged. Joint-space is better in standard settings but task-space is better for transfer/cross-embodiment. This makes the practical takeaway less crisp and reduces the overall impact of the study. The paper is honest about this, but it remains a limitation.
- Limited task diversity. The four real-world tasks are all relatively simple rigid-body manipulation tasks (touch, pick, place, transfer). The absence of contact-rich, deformable-object, or dynamic tasks limits the generalizability of the findings. The simulation tasks add breadth but still fall within a similar manipulation regime.
- Confounded comparison in main results. In Table 1 (the central result), delta actions use execution horizon $k=30$ while absolute actions use $k=60$. While this is motivated by the horizon analysis in RQ1, it means the main comparison does not hold the execution horizon constant, introducing a confound. The paper should more prominently acknowledge this and ideally include a matched-horizon comparison as a reference point.
- Thin foundation model experiments. The π0 transfer learning experiments (Section 4.3.2) involve only three tasks on one platform with LoRA fine-tuning. Given that the paper motivates itself partly through the lens of foundation models and cross-embodiment transfer, this section feels underdeveloped relative to its prominence in the narrative.
- Missing discussion of rotation representations. The choice of rotation parameterization (Euler angles, quaternions, 6D rotation, axis-angle) is a known source of discontinuities and learning difficulty in task-space control, yet it receives no attention. This could significantly interact with the spatial axis findings.
- Limited novelty in methodology. The paper proposes no new methods, architectures, or algorithms. Its contribution is purely empirical, synthesizing known design choices. While this is valuable, it somewhat limits the lasting impact — the conclusions could shift with the next generation of policy architectures.

---

> ### Author Rebuttal · Authors · 2026-03-31
>
> Thank you for the constructive reviews. We address each concern below. Additional experimental details and demos are available in [this url](https://sites.google.com/view/icml-2026-rebuttal).
>
> > W1: Inconclusive spatial axis results
>
> Yes, the spatial axis conclusions are more nuanced than the temporal axis, but this reflects the **inherent nature** of these representations. Uncovering and formalizing these nuanced but consistent boundaries is a core contribution of this study.
>
> > W2: Limited task diversity
>
> We added **cloth folding**, a deformable and more spatially demanding task. For evaluation, we report success rates over 5 trials for the easy setting and 3 trials for the hard setting. The results remain consistent with our main conclusions. Please see the above url for detailed video results.
>
> |          Setting         | Delta     EEF     |       Delta Joint          | Absolute  EEF    |    Absolute Joint    |
> | ------------------ | --------------- | --------------- | ------------- | ----- |
> | w/o Pretrain  (Easy) | 2/5            | **3/5** | 1/5          | 2/5  |
> | w/o Pretrain  (Hard) | 0/3             | **1/3**   | 0/3           | 0/3   |
> | w Pretrain (Easy) | **5/5** | **5/5** | 4/5       | 4/5  |
> | w Pretrain (Hard) | **3/3**   | **2/3**   | **2/3** | 2/3   |
>
> > W3 & Q1: Horizon confound in Table 1
>
> Thanks for the advice, here we provide a matched-horizon comparison. These results are consistent with our findings and we will include these results in the revision.
>
> | Horizon  | Delta EEF | Delta Joint | Abs EEF   | Abs Joint |
> | ---------------- | --------- | ----------- | --------- | --------- |
> | **k = 30** | `71.0` | **`76.7`**   | `56.8` | `59.3` |
> | **k = 60** | `64.5` | **`68.3`**   | `64.3` | `67.5` |
>
> > W4: Thin foundation model experiments
>
> Scaling experiments to all 14 tasks and configurations with fully tuned large VLA models like $\pi_0$ was beyond our compute resource. However, to further strengthen this claim, we have included additional foundation model experiments in the **cloth folding** task (see W2 table, "w Pretrain" rows) using the X-VLA pretrain weights. These new results further confirming that task-space representations become increasingly competitive in the foundation model era.
>
> > W5 & Q2: Rotation representation
>
> We standardized on 6D rotation for all task-space tasks due to its superior stability for policy learning. This choice avoids the inherent discontinuities and ambiguities of Euler or quaternion formats, ensuring that the rotation representation itself does not act as a confounder in our evaluation of task-space control.
>
> > W6: Limited novelty in methodology
>
> Our contribution is providing the first comprehensive, large-scale, controlled empirical consolidation that demystifies how action design choices interact with learning dynamics. We believe the identified properties offer valuable insights that remain relevant even as policy architectures evolve.
>
> > Q3: IK solver sensitivity
>
> We used the default IPOPT-based numerical IK solver provided by AgileX. Further, we evaluated **3 additional IK configurations** with different optimization steps, thresholds, and smoothness coefficients. These changes slightly affect the success rates and they do **not** alter the relative ordering or the consistency of our conclusions.
>
> | (Steps,Threshold,Smooth) | Delta EEF | Delta Joint | Abs EEF   | Abs Joint |
> | --------- | --------- | ----------- | --------- | --------- |
> | 10, $10^{-2}$, 0  | `86.6` | **`93.8`**   | `68.7` | `86.7`|
> | 200, $10^{-4}$, 0   | `89.9` | **`93.8`**   | `74.3` | `86.7` |
> | 400, $10^{-4}$, 0.1   | `89.1` | **`93.8`**   | `69.6` | `86.7` |
>
>
> > Q4: Kinematic similarity of AgileX and AIRBOT
>
> No, they are not kinematically equivalent. We summarize several representative geometric and joint-range differences below:
>
> |  | AgileX | AIRBOT |
> | :--- | :--- | :--- |
> | Max Reach | 626 mm | 647 mm |
> | Link Length Ratio | 1.14 : 1 | 0.91 : 1 |
> | J1 Working Range | ±154° | ±170° |
>
> The performance gap confirms this morphological mismatch. Joint-space policies fail to transfer due to misaligned configuration spaces, whereas task-space policies succeed by unifying these differences in a Cartesian frame.
>
> > Q5: Spatial variation in performance
>
> Yes. We observe systematic spatial variation across the workspace grid and the magnitude of this variation is representation-dependent:
>
> | Metric | Delta EEF | Delta Joint | Abs EEF | Abs Joint |
> | ------ | --------- | ----------- | ------- | --------- |
> | Success std. across spatial locations | `0.0580` | `0.0643` | `0.0782` | `0.0862` |
>
> These results show that delta consistently yields lower spatial variance than absolute, and within each temporal setting EEF is slightly more uniform than joint-space, likely because Cartesian actions align more directly with workspace geometry while joint is affected by kinematic non-uniformity near boundaries and singular regions. We further provide per-cell heatmaps in the demo webpage.

---

> > ### Author Rebuttal · Reviewer_s81P · 2026-04-01
> >
> > I thank the authors for their thorough and responsive rebuttal.
> >
> > The matched-horizon comparison (W3/Q1) was my primary concern, and the new results convincingly demonstrate that the delta advantage holds at both $k=30$ and $k=60$, effectively resolving the confound. The IK solver ablation (Q3) and spatial variance analysis (Q5) add valuable robustness checks that strengthen the paper's claims. The kinematic comparison between AgileX and AIRBOT (Q4) adequately addresses the morphological similarity concern. The cloth folding experiment (W2), while limited in trial count, provides directional evidence for generalization to deformable tasks.
> >
> > I strongly encourage the authors to incorporate the matched-horizon comparison and the spatial variance analysis into the main paper in the revision, as these results meaningfully strengthen the central narrative.
> >
> > Given the new evidence provided, I affirm my score of 5 (Accept).
> >
> > I would like to congratulate the authors for submitting this important work. In the same spirit as that of Reviewer 59CY's comment, I believe this kind of work is precisely what the community needs to advance our understanding of *why* and *how* modern robot learning techniques work, which is indeed relevant to the ML community.

---

> > > ### Author Response · Authors · 2026-04-06
> > >
> > > We sincerely thank the reviewer for the detailed feedback and thoughtful evaluation. We are glad that the additional experiments have addressed the concerns and strengthened the paper. We will ensure these results are included in the final version of the paper.

---

### Official Review · Reviewer_iES3 · 2026-03-14

**Soundness:** 2
**Presentation:** 3
**Significance:** 3
**Originality:** 3
**Overall Recommendation:** 4
**Confidence:** 4

**Summary:**

The paper presents a large-scale empirical study of action space design for robotic manipulation policies, systematically analyzing how temporal representations and spatial parameterizations affect learning stability, performance, and generalization.

**Compliance With Llm Reviewing Policy:**

Affirmed.

**Final Justification:**

Just raised my score from 3 to 4. The rebuttal meaningfully improves confidence; the conclusions are now well-supported across multiple settings.

**Key Questions For Authors:**

Do the authors expect the same conclusions to hold for high-DoF end-effectors (e.g., multi-finger hands)? How about other embodiments/platforms (e.g., other arms, humanoids) or other families of manipulation tasks (e.g., dynamic manipulation, contact-rich manipulation, deformable object manipulation)? Why or why not?

How sensitive are the results to the specific low-level controller used in the robot hardware?

Would the same trends hold for other architectures such as reinforcement-learning policies or diffusion transformers?

Can the authors provide deeper theoretical justification for why delta actions improve learning beyond empirical observations?

**Limitations:**

Yes

**Strengths And Weaknesses:**

Strengths:

- The manuscript considers important empirical questions related to action space design

- High-quality empirical work; 13,000+ real-world robot rollouts is a substantial experimental effort

- The paper is well-written, with clear presentation of the manuscript’s claimed assertions/takeaways

Weaknesses:

- General remark: Even with noise amplification analysis, the paper’s main contribution is largely empirical rather than methodological. While valuable for practitioners, the manuscript would likely be much better-received at RSS and maybe CoRL, compared to ICML.

- Section 3: The real-world tasks are relatively simple: touch cube, pick cup, pick-and-place, and bimanual transfer. These tasks may not fully represent the complexity of modern manipulation benchmarks.

- Section 3: Even with the manuscript’s systematic experiments, several factors could influence results, including controller gains, IK solver stability, data formats and action normalization, low-level control loop efficiency, etc. The paper makes strong causal assertions, but some of these variables are not fully controlled.

- Section 4, Table 1: The conclusions and results depend on a specific robot hardware; most of the real-world experiments use AgileX platform. It is unclear whether the results generalize to high-DoF robot hands, other arms, humanoids, dynamic manipulation tasks, contact-rich manipulation tasks, deformable object manipulation tasks, etc.

- Section 4: The observed superiority of joint-space actions may reflect limitations of the inverse kinematics solver rather than intrinsic properties of the action representation.

---

> ### Author Rebuttal · Authors · 2026-03-31
>
> Thank you for the constructive feedback. We address each concern below. Additional details, demos and visualizations are available in [this url](https://sites.google.com/view/icml-2026-rebuttal).
> > W1: Better suited for RSS/CoRL than ICML
>
> We respectfully believe ICML is an appropriate venue: as end-to-end robot learning becomes a core ML problem, insights on action representation choices is valuable beyond robotics. More importantly, our contribution is not only empirical scale, but also a **systematic analysis** of how temporal and spatial abstractions affect optimization and generalization.
>
> > W2: Real-world tasks are relatively simple
>
> We added **cloth folding**, a deformable and more spatially demanding task. For evaluation, we report success rates over 5 independent trials for the easy setting and 3 trials for the hard setting. The results remain consistent with our main conclusions. Please see the above url for detailed video results.
>
> | Setting             | Delta EEF | Delta Joint | Absolute EEF | Absolute Joint |
> | ------------------- | --------- | ----------- | ------------ | -------------- |
> | w/o Pretrain (Easy) | 2/5       | **3/5**     | 1/5          | *2/5*          |
> | w/o Pretrain (Hard) | 0/3       | **1/3**     | 0/3          | 0/3            |
> | w Pretrain (Easy)   | **5/5**   | **5/5**     | *4/5*        | *4/5*          |
> | w Pretrain (Hard)   | **3/3**   | **2/3**     | **2/3**      | *2/3*          |
>
> > W3: Confounding factors not fully controlled
>
>
> Real-world robot learning is a highly coupled system with many factors that can affect results. To keep the comparison credible, we isolate the factors as much as possible from a learning perspective, while keeping the low-level system design aligned with **community practice**. Concretely, the low-level controllers, IK solvers, and data formats follow the officially provided standard implementations, rather than a specially engineered setup. We believe this makes the current setting transparent and more useful for practical guidance. To further strengthen the evidence, we additionally include **representative robustness checks** on selected system components (hyperparameters for IK sovlers). These choices may affect the **absolute numbers**, but they do not change the **conclusions**:
>
>
> | Config (Steps, Threshold, Smooth) | Delta EEF | Delta Joint | Abs EEF   | Abs Joint |
> | --------- | --------- | ----------- | --------- | --------- |
> | 10, $10^{-2}$, 0  | `86.6` | **`93.8`**   | `68.7` | `86.7`|
> | 200, $10^{-4}$, 0   | `89.9` | **`93.8`**   | `74.3` | `86.7` |
> | 400, $10^{-4}$, 0.1   | `89.1` | **`93.8`**   | `69.6` | `86.7` |
>
>
>
> > W4: unclear generalization to other embodiments and tasks
>
> We added **cloth folding** task (see W2). More broadly, the paper already spans **single-arm**, **bimanual**, **cross-embodiment transfer** to AIRBOT, and 10 RoboTwin-2.0 tasks, providing evidence across substantially different settings.
>
>
> > W5: Joint-space superiority may reflect IK solver limitations.
>
> No. Joint-space superiority is strongest under **flow matching**, suggesting a representation/model-capacity effect rather than an IK artifact. Moreover, in cross-embodiment transfer, **task-space outperforms joint-space**, and our IK ablation changes absolute success rates but not the relative ordering.
>
> > Q1: more embodiments and task?
>
> Our findings already extend to a deformable task (cloth folding) and to kinematically distinct platforms (AgileX and AIRBOT). We suggest that high-DoF systems such as dexterous hands or humanoids are a different regime, where hierarchical or hybrid action spaces may be more appropriate.
>
> > Q2: Sensitivity to low-level controller?
>
> The consistency across multiple platforms suggests the findings are not tied to a single controller. The new ablations in W3 further show that controller-related changes affect **absolute values** but preserve the **relative conclusions**.
>
> > Q3: Same trends..?
>
> Yes. The main paper already includes a foundation model (**pi0**), and the new cloth-folding study adds a **DiT-based VLA** setup. The same qualitative trends remain consistent.
>
> > Q4: Theoretical justification?
>
> Yes. As discussed in Appendix G.2, **absolute** actions can be harder to learn because the policy must infer global geometry and localization from observations. To provide further insight in the rebuttal, we additionally analyze the action labels derived from the expert trajectories. We find that delta actions exhibit lower cross-location drift (Delta EEF `2.924` vs. Abs EEF `4.986`; Delta Joint `2.680` vs. Abs Joint `4.163`) and substantially weaker dependence on workspace position (Delta EEF `0.171` vs. Abs EEF `0.620`; Delta Joint `0.124` vs. Abs Joint `0.590`). These additional analyses further support why delta representations are easier to learn in practice, and we provide the detailed analysis on the demo website.

---

> > ### Author Rebuttal · Reviewer_iES3 · 2026-04-05
> >
> > Thanks for the detailed rebuttal and additional experiments; my concerns are largely resolved. Increasing to 4.

---

> > > ### Author Response · Authors · 2026-04-06
> > >
> > > We thank the reviewer for the positive feedback. We will further incorporate these results and related discussions into the revision to improve clarity and completeness.

---

### Decision · Program_Chairs · 2026-04-30

**Decision:**

Accept (regular)

**Comment:**

This paper presents a large-scale empirical study on action space design for robotic manipulation, backed by over 13,000 real-world rollouts. The reviewers recognized the scale and rigor of the evaluation, and all concerns were fully resolved after the rebuttal, with every reviewer raising their score.